# Large Language Models as Narrative-Driven Recommenders

## Abstract

Narrative-driven recommenders aim to provide personalized suggestions for user requests expressed in free-form text such as "*I want to watch a thriller with a mind-bending story, like Shutter Island.*" Although large language models (LLMs) have been shown to excel in processing general natural language queries, their effectiveness for handling such recommendation requests remains relatively unexplored. To close this gap, we compare the performance of 38 open- and closed-source LLMs of various sizes, such as LLama 3.2 and GPT-4o, in a movie recommendation setting. For this, we utilize a gold-standard, crowdworker-annotated dataset of posts from reddit's movie suggestion community and employ various prompting strategies, including zero-shot, identity, and few-shot prompting. Our findings demonstrate the ability of LLMs to generate contextually relevant movie recommendations, significantly outperforming other state-of-the-art approaches, such as doc2vec. While we find that closed-source and large-parameterized models generally perform best, medium-sized open-source models remain competitive, being only slightly outperformed by their more computationally expensive counterparts. Furthermore, we observe no significant differences across prompting strategies for most models, underscoring the effectiveness of simple approaches such as zero-shot prompting for narrative-driven recommendations. Overall, this work offers valuable insights for recommender system researchers as well as practitioners aiming to integrate LLMs into real-world recommendation tools.

## CCS Concepts

• **Information systems → Recommender systems**.

## Keywords

Recommender systems, Large language models, Narrative-driven recommendations, Movie recommendations, Prompting strategies

**ACM Reference Format:**
Anonymous Author(s). 2025. Large Language Models as Narrative-Driven Recommenders. In *Proceedings of Proceedings of the ACM Web Conference 2025 (WWW '25)*. ACM, New York, NY, USA, 19 pages. https://doi.org/XXXXXXX.XXXXXXX

## 1 Introduction

Recently, tools powered by large language models (LLMs), such as OpenAI's chatbot ChatGPT, gained attention due to their high performance in various natural language processing (NLP) tasks.

For example, LLMs showed increased performance on tasks such as translation, question-answering, cloze tests [9], linguistic analyses of generated content [20], or re-ranking tasks [51]. In some initial studies related to recommender systems, LLMs also demonstrated great potential for specific recommendation tasks, such as explanation [33], ranking [11], as well as conversational [22], content-based [32], or next-item recommendations [54]. However, suitability of LLMs for a particularly promising narrative-driven recommendation scenario [8] remains still relatively unexplored.

In narrative-driven recommendation scenarios users pose free-form requests such as "*I just want to see a movie where the good guy kicks some ass!*" These queries are commonly submitted to online forums such as reddit, where communities respond with tailored suggestions (Fig. 1, right). While asynchronous community responses to such posts generally fulfill users' requests, recent studies applying traditional machine learning methods to such recommendation scenarios indicate potential for improvement, as the problem of narrative-driven recommendations proves difficult for existing recommender approaches [14–16]. Given these challenges, in this paper, we set out to evaluate the potential of LLMs in such a recommendation scenario, leveraging their capabilities to better understand and process user-generated narratives.

To this end, we probe LLMs with movie recommendation requests originally posed by real users to the r/MovieSuggestions[1] community on reddit (Fig. 1, center). Utilizing these user requests along with the accompanying comments that contain community recommendations [16], we investigate how well responses by open- and closed-source LLMs match the recommendations from the reddit community. In total, we evaluate 38 state-of-the-art LLMs, categorized by size, ranging from tiny (<4 billion parameters) to large (≥50 billion parameters), employing various prompting strategies. Specifically, we assess the performance of these models using zero-shot [29], identity [28], and few-shot [17, 18] prompting (Fig. 1, left). Finally, we compare the performance of the evaluated LLMs with other recommendation approaches, such as doc2vec [30].

Our results demonstrate that LLMs can effectively respond to natural language prompts, translating user requests into relevant recommendations. We present four key findings from our comprehensive evaluation of movie recommendation tasks. First, we observe that using basic zero-shot prompting, LLMs are able to generate movie recommendations that rival or surpass those of traditional and state-of-the-art recommender approaches. In particular, GPT-4o as the overall best-performing LLM exhibits a recommendation performance 70% higher than the baseline. Second, expanding the prompting techniques through identity or few-shot prompting does not significantly improve recommendation performance, underscoring the strength of LLMs when using simple zero-shot prompting for this task. Third, we find that medium-sized open-source models (e.g., Gemma 2 27B) perform competitively with similarly sized closed-source models (e.g., GPT-3.5 Turbo) and even larger open-source models (e.g., Mistral Large 2 123B). Finally, our

---

[1]https://www.reddit.com/r/MovieSuggestions/

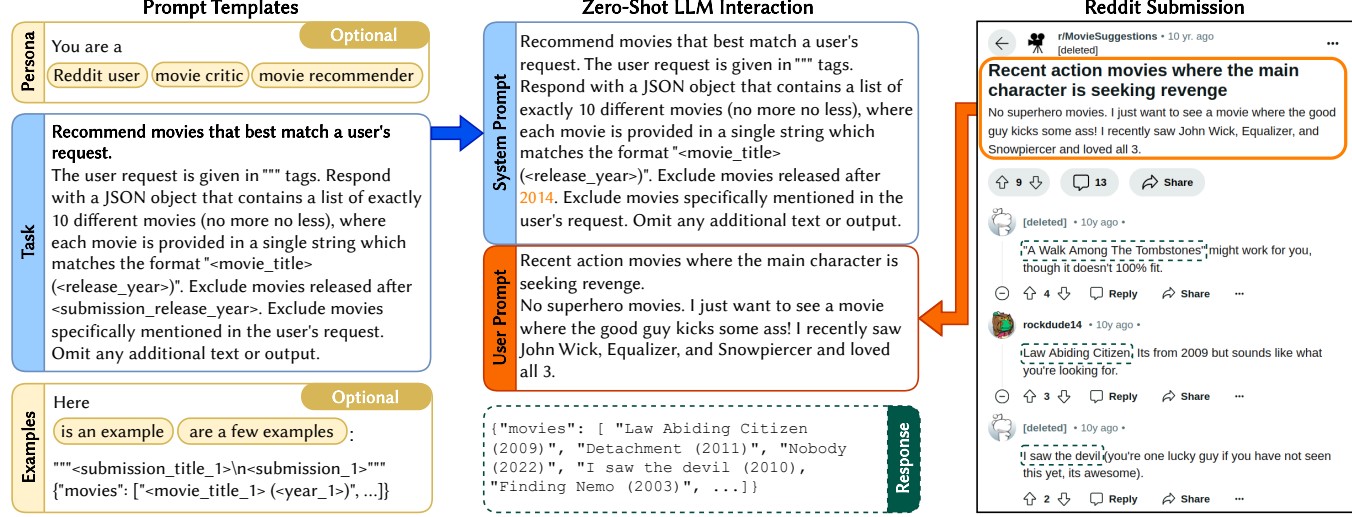

**Figure 1: Evaluation of Movie Recommendations Using LLMs and Reddit Submissions.** We assess LLM movie recommendation by combining different prompt templates (left) into zero-shot (*Task*), identity (*Persona* and *Task*), and few-shot (*Task* and *Examples*) prompting. We utilize these different prompts* to produce suggestions for real movie requests submitted by reddit users (right) by generating these recommendations with various LLMs (center). The LLM-generated recommendations (bottom center, dashed box) are compared to actual responses from reddit's community (right, dashed boxes) to evaluate the performance of LLMs as narrative-driven movie recommenders.

*We instruct the LLMs to exclude movies released after the year of the reddit request, to enforce that responses match our use case.

findings hold even when accounting for potential data leakage, specifically the possibility that data used for our evaluation may have also been incorporated during the pre-training phase of certain LLMs. Altogether, this work offers practical insights for both recommender system researchers and practitioners looking to integrate LLMs into real-world recommendation applications. Finally, we make our code and employed datasets publicly available.[2]

## 2 LLMs as Recommender Systems

### 2.1 Setup

**Dataset.** We use a crowdworker-curated dataset[3] for our experiments [16]. This dataset consists of annotated submissions and comments from reddit [7], more specifically from the subreddit r/MovieSuggestions. In particular, this dataset includes reddit submission IDs, titles, and original texts, as well as movie titles, actor names, keywords, and genres that were annotated by crowdworkers. Each recommendation request in the dataset contains one or more positively mentioned movies (i.e., examples of movies that the user liked before) as well as additional information, such as negatively mentioned movies (i.e., movies that the user did not like before), positively or negatively mentioned keywords (describing further aspects of the movies), and genres. While the full dataset contains 1 480 submissions (from August 2011 to July 2017) making up test and training data, for fair comparison with prior work [14] we solely utilize the test set of 296 submissions (from November 2016 to July 2017), with 778 unique mentioned movies and 4 329

comments including over 11 000 individual recommendations by reddit users and 3 593 different movies as suggestions.

**LLMs.** In our experiments, we use 35 open-source and three commercial OpenAI LLMs, spanning 13 distinct model families (Table 1). In particular, we use Ollama[4] with q4_0 quantization [25] for all open-source LLMs and OpenAI's paid API[5] for the closed-source GPT models. We group the models into four categories based on the number of parameters: tiny (<4 billion), small ($\geq$4 and <10 billion), medium ($\geq$10 and <50 billion), and large ($\geq$50 billion) LLMs. In this paper, we focus on the evaluation of prompting strategies and do not optimize fine-grained configuration of model parameters, such as temperature [43]. However, we change the context window size to 4 096 tokens (Ollama default is 2 048) to avoid truncation of longer LLM requests. We also set the maximum number of tokens for response generation to 500 to enable reasonable response times.

### 2.2 Prompting Experiments

We evaluate three popular prompting strategies including zero-shot, identity, and few-shot prompting in separate experiments. In particular, for each reddit submission from the test dataset we generate a single LLM request that is composed of a system prompt and a user prompt (Fig. 1, center). While the user prompt maintains a consistent format across all experiments—comprising the title and body of a submission within specified tags—the system prompt varies depending on the experiment. The system prompt includes a *Task* section and, depending on the experiment, optional *Persona* and *Examples* sections (Fig. 1, left).

---

[2]https://osf.io/uc6r2/?view_only=813aec647d7344a7ab782da2978fb7dc
[3]https://doi.org/10.17605/osf.io/ma2bj

[4]https://ollama.com
[5]https://platform.openai.com

**Zero-Shot Prompting.** In this experiment, we assess out-of-the-box performance of LLMs as narrative-driven recommenders in the movie domain. Hence, our zero-shot prompt consists of a *Task* section including instructions for the model to recommend movies based on a user's specific request provided in the form of tags, and constraints that define the format specifications and limitations for the expected output. To facilitate post-processing, we

Table 1: Evaluated LLMs. We report size category, family, name, number of parameters, and relevant references for all 38 investigated models. We always utilize the newest model versions as of June to September 2024.

| Model Category | Model Family | Model Name | Params. ($10^9$) | Ref. |
|---|---|---|---|---|
| Tiny | Gemma | Gemma 2B | 2.51 | [52] |
| | Gemma 2 | Gemma 2 2B | 2.61 | [19] |
| | Llama 3.2 | Llama 3.2 1B | 1.24 | [36] |
| | Llama 3.2 | Llama 3.2 3B | 3.21 | [36] |
| | Phi-3 | Phi-3 3.8B | 3.82 | [1] |
| | Phi-3.5 | Phi-3.5 3.8B | 3.82 | [38] |
| | Qwen2 | Qwen2 0.5B | 0.49 | [57] |
| | Qwen2 | Qwen2 1.5B | 1.54 | [57] |
| | Qwen2.5 | Qwen2.5 0.5B | 0.49 | [53] |
| | Qwen2.5 | Qwen2.5 1.5B | 1.54 | [53] |
| | Qwen2.5 | Qwen2.5 3B | 3.09 | [53] |
| Small | Gemma | Gemma 7B | 8.54 | [52] |
| | Gemma 2 | Gemma 2 9B | 9.24 | [19] |
| | GPT | GPT-4o mini | * | [39] |
| | Llama 3 | Llama 3 8B | 8.03 | [35] |
| | Llama 3.1 | Llama 3.1 8B | 8.03 | [34] |
| | Mistral | Mistral 7B | 7.25 | [26] |
| | Qwen2 | Qwen2 7B | 7.62 | [57] |
| | Qwen2.5 | Qwen2.5 7B | 7.62 | [53] |
| | Yi | Yi 6B | 6.06 | [2] |
| | Yi | Yi 9B | 8.83 | [2] |
| Medium | Gemma 2 | Gemma 2 27B | 27.2 | [19] |
| | GPT | GPT-3.5 Turbo | * | [41] |
| | Mistral | Mistral NeMo 12B | 12.2 | [6] |
| | Mistral | Mistral Small 22B | 22.2 | [3] |
| | Mixtral | Mixtral 8x7B | 46.7 | [27] |
| | Phi-3 | Phi-3 14B | 14 | [1] |
| | Qwen2.5 | Qwen2.5 14B | 14.8 | [53] |
| | Qwen2.5 | Qwen2.5 32B | 32.8 | [53] |
| | Yi | Yi 34B | 34.4 | [2] |
| Large | GPT | GPT-4o | * | [40] |
| | Llama 3 | Llama 3 70B | 70.6 | [35] |
| | Llama 3.1 | Llama 3.1 70B | 70.6 | [34] |
| | Llama 3.1 | Llama 3.1 405B | 406 | [34] |
| | Mistral | Mistral Large 123B | 123 | [5] |
| | Mixtral | Mixtral 8x22B | 141 | [4] |
| | Qwen2 | Qwen2 72B | 72.7 | [57] |
| | Qwen2.5 | Qwen2.5 72B | 72.7 | [53] |

*We estimate the size of closed-source models to be within the bounds of the corresponding model category.

request the model to return a JSON object containing exactly ten movie recommendations to calculate our evaluation metrics @10 (e.g., F1@10)—for fair comparisons with previous results [14]. To simplify the distinct mapping of movies during evaluation, we request from LLMs to format each recommendation as a single string including the movie's title and release year (e.g., "Titanic (1997)"). Additionally, to compare recommendations with the gold-standard recommendations from the reddit community, we request that recommended movies are released before the date of the original reddit submission. Finally, the zero-shot prompt serves as the foundation that we extend in the remaining two experiments.

**Identity Prompting.** The *Persona* section of the prompt templates defines the LLM identity such as a reddit user, a movie critic, or a movie recommender. Hence, in the identity prompting experiments, this section is prefixed to the system prompt used in the zero-shot experiment to direct LLM responses towards a particular persona.

**Few-Shot Prompting.** In few-shot prompting, we provide multiple random input-output examples that showcase the requested response structure. Thus, we extend the zero-shot system prompt with the *Examples* section containing either one, five, or ten examples to steer the LLM's generating process. These examples contain exemplary user prompts as well as JSON responses.

## 2.3 Evaluation of LLM Responses

Following our prompting experiments, we parse the corresponding LLM responses, match the contained titles to actual movies, and compute established evaluation metrics to assess LLM performance.

**Identifying Recommendations.** After prompting the LLMs, we proceed by extracting recommendations from the JSON responses. In particular, we apply a regex to parse correct recommendations in the format "`<movie_title> (<release_year>)`" (Fig. 1, center) and remove any duplicate strings as well as ignoring case. Finally, for responses containing more than ten movies, we randomly sample ten to compute our evaluation metrics.

**Title Matching.** We compare LLM recommendations to the recommendations from the reddit community by exact matching of title and release year. As the community recommendations in the dataset are identified by their IDs on IMDb, we first filter out recommendations that were already mentioned in the submission text. Then, we retrieve the titles of the remaining community recommendations from IMDb in all available languages (e.g., original title: "Un prophète (2009)", international title: "A Prophet (2009)"). Additionally, we apply soft matching using Ratcliff/Obershelp pattern-matching [46] with a threshold of 0.9. As we observe no significant difference in the results between exact and soft matching, we only report the exact matching results.

**Evaluation Metrics.** We compute four standard recommendation metrics—precision, recall, F1 score, and Normalized Discounted Cumulative Gain (NDCG) [44, 59], with fixed cut-off value (i.e., Precision@10, Recall@10, F1@10, NDCG@10[6])—for each request and report their overall means over the whole test dataset (i.e., macro average). Due to space limitation, we report results aggregated by model size category and prompting strategy for all metrics and only

---

[6]Note that Recall@10 and F1@10 have an upper limit of 0.43 and 0.57, resp., as the number of community suggestions varies per submission (mean=30.77).



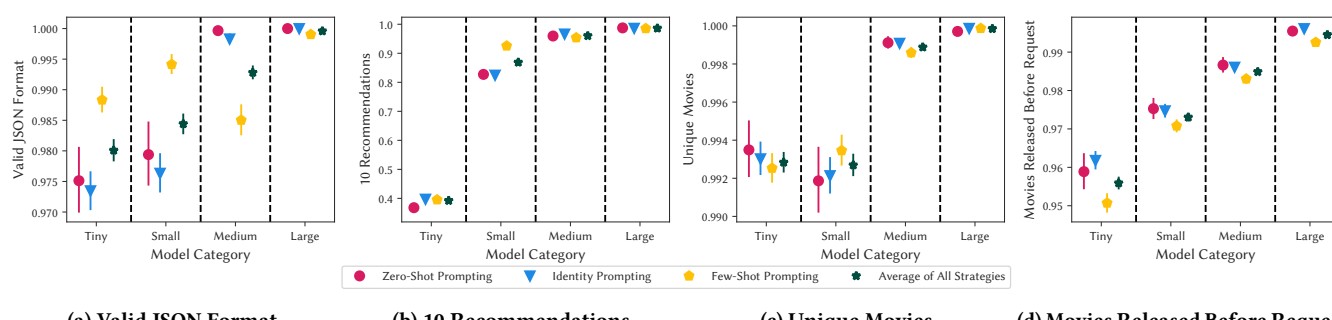

(a) Valid JSON Format    (b) 10 Recommendations    (c) Unique Movies    (d) Movies Released Before Request

**Figure 2: LLM Response.** We report the ratio of (a) LLM responses with valid JSON, (b) responses with exactly ten recommendations, (c) unique movies, and (d) movies released before request, with vertical bars showing bootstrapped 95% confidence intervals (often too narrow to be visible). We observe that medium and large LLMs mostly follow the constraints outlined in our prompt (Fig. 1, left), while tiny and small LLMs show slightly lower reliability, though their overall numbers remain high.

F1 results for individual models in the main text. We present more detailed results for all evaluation metrics in the Appendix.

**Response Variance.** To estimate the variance of LLM responses, we repeat each request 30 times for all models in the zero-shot prompting experiment and compute one-way ANOVA. The test results show no significant differences across repetitions in any of our evaluation metrics and models (Appendix, Table 2). For this reason, we repeat all other experiments only three times and use the repetition with the median F1 score as the final result.

## 3 Results

We present our experimental findings across 38 evaluated LLMs. First, we assess the structural correctness and validity of the LLM responses in relation to the constraints posed in our prompts. We then evaluate the performance of LLM recommendation, especially considering different-sized LLMs, the effectiveness of different prompting strategies, and open- versus closed-source models. Finally, we compare LLMs to other recommendation approaches and present a robustness experiment regarding potential data leakage. To statistically substantiate our results, we bootstrap the dataset in all experiments and report bootstrapped 95% confidence intervals.

### 3.1 Format Adherence

We assess the compliance of the LLM-generated outputs with the formatting and structural instructions by checking correctness of the JSON format, the total number of returned recommendations, the frequency of unique movie entries within a recommendation list, and the movie release year.

**JSON Format.** Our analysis of LLM responses reveals that the fraction of valid JSON responses exceeds 97% for all aggregations of model size category and prompting strategy (Fig. 2a). This demonstrates high proficiency of LLMs in generating correctly formatted responses. Large LLMs exhibit exceptional performance, achieving valid JSON output in over 99.9% of cases across all prompting strategies. In more details (Appendix, Fig. 8), the GPT models as well as Gemma 2 9B, Llama 3 70B, Llama 3.1 8B and 70B, and Qwen 2.5 14B consistently produce 100% valid JSON responses. In contrast, some of the smaller models are more prone to error, such as Llama 3.2 1B with 93.39%, Mistral 7B with 91.75%, Qwen 2 0.5B with 92.76%,

Yi 6B with 94.45%, and Phi-3 14B with around 94% of correct JSON responses. Moreover, for Phi-3 14B we observe a significant decline in performance for few-shot prompting, likely due to the model's difficulty in processing longer requests.

**Number of Recommendations.** Compliance with the constraint of responding with exactly ten recommendations varies considerably across model size categories and prompting strategies (Fig. 2b). Large LLMs reach nearly perfect adherence to the prompt, with the fraction of responses containing precisely ten recommendations exceeding 98% on average over all models and prompting strategies. Medium-sized models attain an average score of 96% over all prompting strategies with slightly greater variability than their larger counterparts. In contrast, smaller, especially tiny-sized models demonstrate more inconsistent behavior. On average, less than 40% of valid JSON responses from tiny models contain exactly ten recommendations. Furthermore, different prompting approaches (e.g., zero-shot versus few-shot) show only marginal and no significant differences in the average number of recommendations, except for small-sized LLMs with a significant higher average when using few-shot prompting. This highlights the robustness of modern LLMs in meeting specific output requirements with simple prompting configurations, particularly for well-trained and larger architectures. In detail, we observe that each large LLM consistently provide exactly ten recommendations in over 91% of the requests, regardless of the prompting strategy (Appendix, Fig. 9). Qwen2 72B achieves the highest adherence to the number of recommendations constraint, with more than 99.9% of its responses meeting this criterion. In contrast, the smaller models in the tiny category, such as Qwen2 and Qwen2.5 0.5B, frequently fail to include any movies in the zero-shot and identity prompting experiments, with over 92% of their responses lacking movie recommendations. However, the performance of these smaller models improve significantly when provided with few-shot prompts, reducing the proportion of responses without any movie recommendations.

**Unique Movies.** On average across all model size categories and prompting strategies, LLMs return over 99% unique movies in their valid outputs (Fig. 2c). However, tiny and small models exhibit a marginal but statistically significant increase in duplicate movie recommendations. Gemma 2B and 7B, alongside Qwen 2.5 7B, show

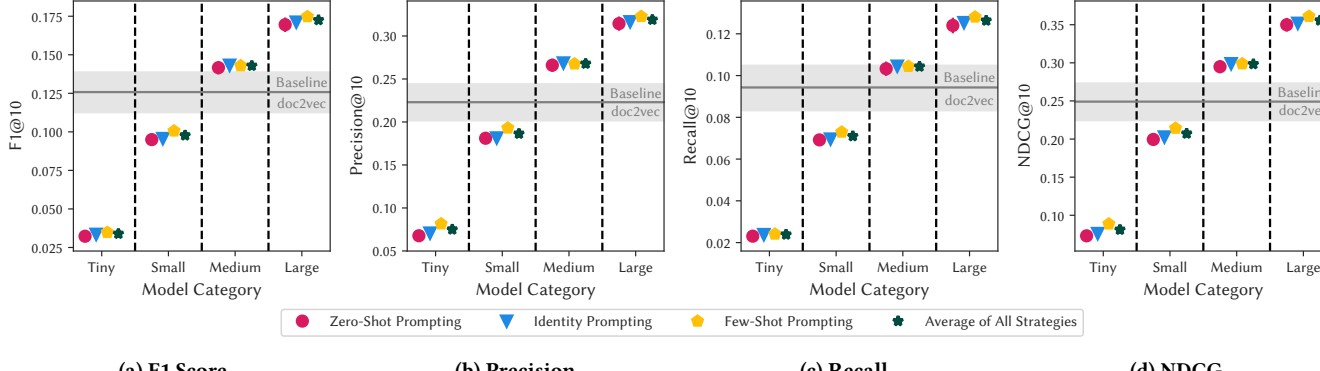

(a) F1 Score                     (b) Precision                     (c) Recall                     (d) NDCG

Figure 3: Aggregated Performance Metrics. We report aggregated recommendation performance of LLMs across model size categories and prompting strategies using the metrics (a) F1@10, (b) Precision@10, (c) Recall@10, and (d) NDCG@10, with vertical bars representing bootstrapped 95% confidence intervals (often too narrow to be visible). We indicate the doc2vec baseline performance as horizontal reference line. On average, large LLMs consistently demonstrate superior performance, surpassing the baseline across all metrics regardless of the prompting strategy. Medium-sized models also outperform the baseline substantially, particularly in Precision@10 and NDCG@10.

the highest rates of duplication, averaging 2–4% duplicate recommendations (Appendix, Fig. 10).

**Release Year.** We show the average fractions of recommended movies that were released before or in the year specified in the system prompt, grouped by model size and prompting strategy (Fig. 2d). Across all model categories from tiny to large, the average fractions of recommendations with correct years exceeds 95%, indicating that LLMs are generally effective at adhering to temporal constraints in narrative-driven movie recommendation tasks. Larger models, on average, demonstrate a higher adherence, achieving accuracy rates exceeding 99% regardless of the prompting strategy. We provide more detailed breakdowns in the Appendix (Fig. 11).

## 3.2 Recommendation Performance

Our evaluation of the recommendation performance of various LLMs reveals several key insights regarding different model sizes and prompting strategies. In particular, we highlight the general suitability of medium and large LLMs as narrative-driven recommenders, the effectiveness of zero-shot prompting, competitiveness of medium- with larger-parameterized LLMs, as well as high performance of open-source models. Figure 3 depicts aggregated performance results over model size categories and Figure 4 shows F1 scores (i.e., F1@10) of all models categorized by model families across various prompting strategies. We list further detailed results for all models in the Appendix (Figs. 12, 13, 14, and 15).

**Medium and Large LLMs Outperform Baselines.** To evaluate the general suitability of LLMs as narrative-driven recommenders, we compare our results with the state-of-the-art recommendation baselines evaluated on the same dataset [14]. The baseline methods utilized a range of established recommendation algorithms, such as doc2vec, collaborative filtering, matrix factorization, or a TF–IDF-based approach, with the doc2vec approach being the best-performing method with an F1 score of 0.1258 [0.1125, 0.1388] (horizontal lines in Figs. 3 and 4).

Our experimental results show that the medium and large model categories significantly outperform the traditional methods across all evaluated configurations regardless of the prompting strategy. Specifically, all of the large, the majority of the medium, as well as some of the small LLMs achieve higher scores across all metrics, indicating a substantial improvement when generating movie recommendations from narrative user queries. For example, apart from most of the medium and large models even small-sized LLMs, such as Gemma 2 7B or GPT-4o mini, outperform the best state-of-the-art recommender method on this dataset, doc2vec, across all applied prompting strategies (Fig. 4 and Appendix, Figs. 12, 13, 14, and 15). In particular, the best-performing LLM, large-sized GPT-4o, with identity prompting and the persona movie critic achieves an F1 score of 0.2157 [0.2009, 0.2302], which is more than 70% higher than the performance of doc2vec.

Given the minor differences in experimental setups between our work and the studies from which we derive our baselines [14, 16], we conduct an additional sensitivity experiment. Notably, the recommender approaches serving as our baselines generated exactly ten unique movie recommendations from a predefined IMDb movie pool of around 12 000 movies. In this sensitivity experiment, we map our LLM recommendations to that same movie pool and filter out all others. If the LLM produces fewer than ten recommendations, we repeat the request until we collect exactly ten movies. For this experiment, we only use GPT-3.5 Turbo—a medium-sized, closed-source LLM that performs strongly in our benchmarks and shows comparable results to similar-sized open-source models (e.g., Gemma 2 27B) as well as other closed-source models of varying sizes (e.g., GPT-4o). Using identity prompting with the reddit user persona, we obtain an F1 score of 0.2348 [0.2189, 0.2508], indicating a further improvement in LLM performance over the baselines.

**Effectiveness of Zero-Shot Prompting.** Our results demonstrate that zero-shot prompting achieves a high level of performance across all model size categories (Fig. 3), suggesting that additional prompting complexity with identity or few-shot prompting yields

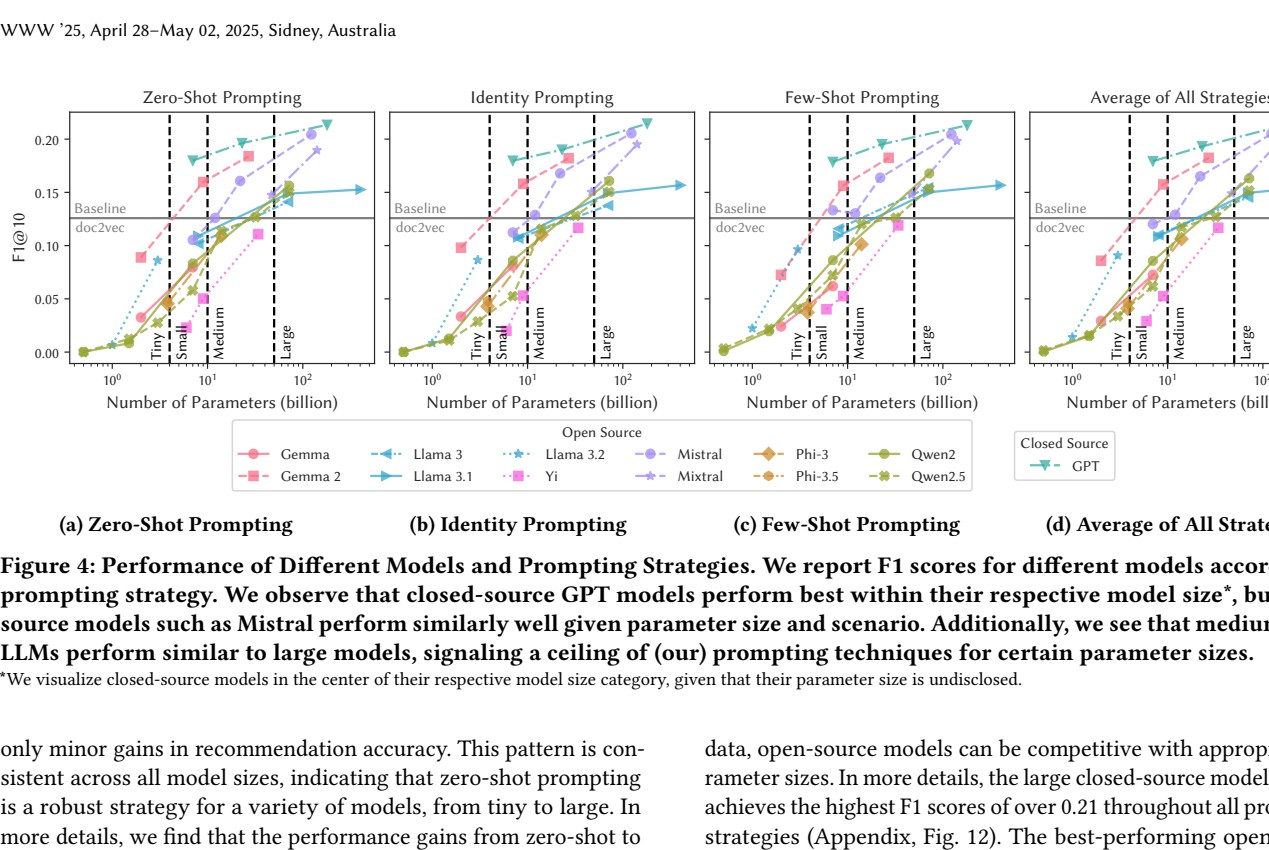

**(a) Zero-Shot Prompting**  **(b) Identity Prompting**  **(c) Few-Shot Prompting**  **(d) Average of All Strategies**

**Figure 4: Performance of Different Models and Prompting Strategies. We report F1 scores for different models according to prompting strategy. We observe that closed-source GPT models perform best within their respective model size\*, but open-source models such as Mistral perform similarly well given parameter size and scenario. Additionally, we see that medium-sized LLMs perform similar to large models, signaling a ceiling of (our) prompting techniques for certain parameter sizes.**

\*We visualize closed-source models in the center of their respective model size category, given that their parameter size is undisclosed.

only minor gains in recommendation accuracy. This pattern is consistent across all model sizes, indicating that zero-shot prompting is a robust strategy for a variety of models, from tiny to large. In more details, we find that the performance gains from zero-shot to identity and few-shot prompting are more substantial for some of the smaller models, suggesting that these strategies are particularly useful for improving the capabilities of less parameterized models (Fig. 4 and Appendix, Figs. 12, 13, 14, and 15). For example, the F1 score using the small-sized Mistral 7B model with zero-shot prompting of 0.1053 [0.0929, 0.1174] significantly increases when applying few-shot prompting up to 0.1342 [0.1219, 0.1462].

**Medium-Sized Models Compete With Larger LLMs.** Despite the general expectation that larger models would significantly outperform smaller ones, our results demonstrate that medium-sized models with parameter counts between 10 and 50 billion can achieve almost equivalent accuracy as their larger counterparts. The aggregated performance scores by model size category and prompting strategy highlight this finding (Fig. 3). Moreover, Figure 4 also illustrates this performance across various models and model families. Model families such as Gemma 2 are competitive with more computationally expensive LLMs such as the larger GPT or Mistral models, achieving similar F1 scores. More detailed results for all models in Appendix (Figs. 12, 13, 14, and 15) further highlight our findings. For example, the best-performing medium-sized GPT-3.5 Turbo model with an F1 score of 0.1932 [0.1876, 0.1986] on average over all prompting strategies, almost performs on par with the large-sized GPT-4o, which exhibits the highest score of all models with an F1 score of 0.2137 [0.2084, 0.2191].

**Closed-Source Over Open-Source Models.** We find that the closed-source GPT models show prominent performances compared to most of the similar sized open-source models within their respective model size categories (Fig. 4). The results further exhibit that medium-sized open-source models often outperform smaller closed-source models, suggesting that even without proprietary

data, open-source models can be competitive with appropriate parameter sizes. In more details, the large closed-source model GPT-4o achieves the highest F1 scores of over 0.21 throughout all prompting strategies (Appendix, Fig. 12). The best-performing open-source model, Mistral Large 2 123B, reaches an F1 score of around 0.20. Medium-sized open-source models such as Gemma 2 27B perform on a similar accuracy level as the closed-source medium-sized GPT-3.5 Turbo model. In the category of small models we find that the closed-source GPT-4o mini slightly but not significantly outperforms the best open-source counterpart, Gemma 2 9B.

### 3.3 Addressing Potential Data Leakage

Given the setup of our experiment, our results may be susceptible to data leakage due to the possible overlap of submissions in our reddit dataset with the LLMs' training data [48]. For this, we follow the methodology of prior work [49] to compare model performance before and after the knowledge cutoff to detect any significant pre- and post-cutoff differences.

**Setup.** As previously, we conduct this robustness analysis with GPT-3.5 Turbo. Given that GPT-3.5 Turbo's knowledge cutoff (i.e., September 2021) is outside the timeframe of our original dataset, we first use GPT-4o as a higher-parameterized expert model to extract recommendations from reddit posts made after this cutoff. Using this dataset, we then repeat our zero-shot experiment with GPT-3.5 Turbo and compare the results to the main zero-shot results.

**Robustness Analysis Dataset.** We consider all submissions from r/MovieSuggestions submitted from October 2021 to July 2024 that contain the word "request" in the submission title to ensure the post contains actual movie inquiries (92 040 submissions). We then keep all submissions with at least five comments with more upvotes than downvotes, focusing on high-quality and engaging user discussions (441 submissions remaining). In contrast to the crowdworker-labeled dataset for our previous experiments, we now employ GPT-4o to extract movie mentions from the requesting user's post as

well as from commenters' replies (Appendix, Fig. 7 for the prompt). Finally, similar as in our original dataset, we exclude submissions with no movies mentioned in the request and less than ten recommendations made by commenters. This final robustness analysis dataset consists of 236 submissions.

**Performance After Knowledge Cutoff.** We rerun our zero-shot prompting experiment using GPT-3.5 Turbo on the robustness analysis dataset, achieving an F1@10 of 0.1667 [0.1513, 0.1814],[7] which does not indicate a significant decline compared to our original results (F1@10: 0.1962 [0.1809, 0.2112]). However, we observe a minor decline in performance, potentially due to movies released after the knowledge cutoff being relevant recommendations for certain requests in the robustness analysis dataset. This supports the validity of our findings, even considering that parts of our evaluation dataset could have been included in the LLMs' training data.

## 4 Discussion

Our findings reveal that LLMs are highly suitable as narrative-driven recommenders. LLMs demonstrate remarkable utility, especially compared to traditional recommender baselines, in generating contextually relevant recommendations from free-form narrative inputs, which can be particularly valuable in enhancing user experience in personalized movie suggestions. In particular, we compare the performance of LLMs as narrative-driven movie recommenders to state-of-the-art recommender approaches [14] and find substantial leaps in recommendation quality. We attribute this to the superior ability of LLMs to interpret complex natural language expressions and nuances, which traditional methods, such as matrix factorization or doc2vec, often fail to capture effectively. Unlike these traditional systems, which rely on extensive feature engineering and domain-specific fine-tuning, LLMs can extract meaningful contextual information directly from narrative requests due to their extensive pre-training.

Moreover, similarly to recent work that demonstrated the effectiveness of zero-shot prompting in various NLP tasks [47, 54], we find that more extensive prompting strategies such as identity or few-shot prompting do not outperform zero-shot prompting. These results suggest that even simple prompting strategies can effectively capture essential user preferences in narrative-driven recommendation tasks, reducing the need for extensive prompt engineering, model training, or fine-tuning. This result aligns with a recent study that evaluated the performance of various prompts for GPT-3.5 Turbo on multiple recommendation scenarios based on Amazon customer reviews, such as rating prediction [31].

Nevertheless, while we find that simple zero-shot prompting performs as well as other strategies such as few-shot prompting, more sophisticated techniques could potentially further enhance recommendation quality. For example, Wang and Lim [54] explored the application of LLMs utilizing GPT-3 for zero-shot next-item recommendation in the context of movie suggestions. The authors developed a three-step prompting strategy to execute subtasks that capture user preferences, select representative previously watched movies, and recommend a ranked list of ten movies. Their experimental evaluation on the MovieLens 100K dataset indicates that

---

[7]Note that Recall@10 and F1@10 now have reduced upper limits of 0.41 and 0.54, resp., due to an increase in community suggestions per submission (mean=37.09).

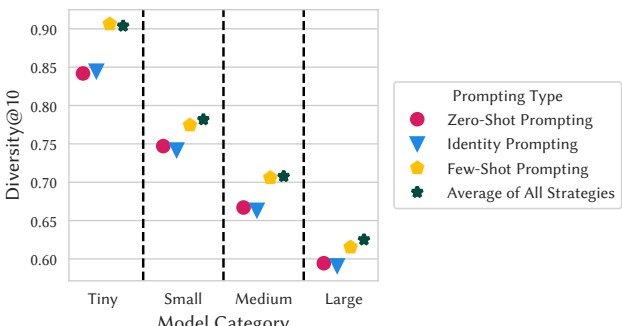

**Figure 5: Inter-List Diversity for All Model Size Categories. We show inter-list diversity with vertical bars indicating bootstrapped 95% confidence intervals (often too narrow to be visible), observing a trend where large LLMs tend to produce less diverse responses across multiple experiment repetitions using the same prompt. Notably, few-shot prompting consistently results in higher diversity across all model sizes.**

this method can outperform traditional recommendation models that require extensive training.

Furthermore, our experiments suggest that medium-sized LLMs, such as Gemma 2 27B, offer a compelling trade-off between computational efficiency and recommendation quality, competing with larger and more resource-intensive models. This finding relates to a study by He et al. [22] in the field of conversational recommenders, where medium-sized models such as GPT-3.5 Turbo produced results comparable to larger counterparts. Thus, medium-sized models emerge as viable alternatives for real-world applications, balancing performance and effectiveness.

Besides, while the closed-source GPT-4o model outperforms all other evaluated LLMs, the minimal performance gap between closed- and open-source models (e.g., Mistral 123B) highlights that open-source LLMs can provide high-quality movie recommendations without the significant computational or cost overhead of closed-source solutions. Liu et al. [32] corroborates this finding, showing that both open- and closed-source LLMs exhibit strong performance in content-based recommendation tasks.

In addition to recommendation performance, we also analyze the inter-list diversity of LLM recommendations (Fig. 5 and Appendix Fig. 16) as diverse recommendations typically enhance user experience in recommender systems [62]. Our investigation reveals that larger models exhibit lower diversity across prompting repetitions, indicating a higher tendency towards stable and repeated outputs. In contrast, smaller models, particularly tiny LLMs, show higher diversity, producing more varied recommendations across iterations. On top of that, few-shot prompting consistently increases recommendation diversity compared to zero-shot or identity prompting, likely due to the inclusion of different random examples across repetitions, as minor variations in prompts often lead to more output variability [60]. Importantly, this increase in diversity does not adversely impact recommendation accuracy across prompting strategies, as demonstrated by our reported results. Therefore, when system operators aim to improve user experience by increasing diversity, few-shot prompting or introducing minor variation in

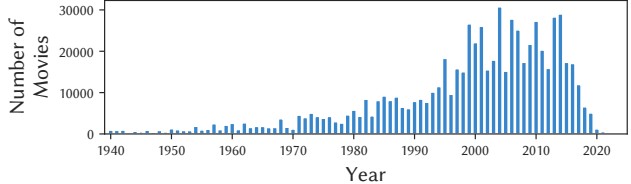

**Figure 6: Release Years. Distribution of movies recommended by LLMs by their release year (visualized from 1940 to 2024) depicts strong preference for movies released after 2000.**

prompts may be preferred over simple zero-shot prompting, as performance remains comparable while allowing for greater variation in the resulting LLM responses.

Finally, as recent works have highlighted the recency bias in LLMs [12], we investigate release years of the recommended movies (Fig. 6 and Appendix, Fig. 17). Our investigation indicates a noticeable preference for more recent movies, particularly those released after 2000. This suggests that LLMs tend to favor newer content, possibly influenced by inherent biases in the training data such as data availability or popularity of contemporary movies.

**Limitations and Future Work.** Our study has certain limitations, which we consider as opportunities for future research. First, our evaluation is restricted to a single domain and dataset from the r/MovieSuggestions subreddit. While this allows for a focused analysis of a wide range of open- and closed-source LLMs across various sizes, our work can be expanded by a broader validation of our approaches in similar recommendation scenarios where users formulate narrative requests for other domains, such as books or music (e.g., on the r/ifyoulikeblank subreddit). Second, we focus on three prompting strategies—zero-shot, identity, and few-shot prompting—while excluding more sophisticated approaches, such as chain-of-thought or tree-of-thought prompting, which have shown promise in tasks requiring strategic lookahead and may enhance recommender adaptability. Additionally, given the marginal differences in the results for our employed strategies, we do not investigate combinations of multiple prompting strategies (e.g., identity and few-shot). Moreover, the integration of multimodal inputs, such as images or audio, alongside textual user queries could further enrich the recommendation process, enhancing the system's ability to respond to complex, multifaceted user preferences. Third, we did not test different individual LLM configurations, including parameters such as sampling method, temperature, or quantization details, which could further modify the output's variability, diversity, and performance. Instead, we employed all LLMs using their default parameter settings, derived from official documentation or Ollama configuration files. When applying our findings to build real-world tools, practitioners should consider further evaluating these default settings to optimize recommendation quality and computational efficiency based on their specific use cases. Finally, we do not explore beyond recency bias to uncover other biases, such as skews in topic or popularity, that may be inherent in LLM responses. Exploring these biases further could yield valuable insights for both recommendation algorithms and pre-training of LLMs.

## 5 Further Related Work

The emergence of LLMs, such as BERT [13] or GPT [45], has led to new developments in NLP. Due to the significant increase in model sizes and the massive amount of training data, LLMs have shown ability of understanding and executing a wide variety of tasks, such as reasoning [10], software engineering [23] or language translation [9]. OpenAI's ChatGPT as well as several competitive open-source LLMs, including Meta's Llama, Mistral, and Microsoft's Phi, have demonstrated exceptional performance, surpassing prior state-of-the-art NLP models across numerous tasks [21, 24, 56, 61].

LLMs are typically instructed through prompts—instructions to translate texts, answer questions, or write essays. Recent advancements in LLMs have sparked growing interest in enhancing their task performance through various prompting strategies [9], such as zero-shot, identity, or few-shot prompting. Further promising prompting strategies, such as chain-of-thought [55] or tree-of-thought [58], enhance the reasoning abilities of generative models, particularly in tasks such as question answering, by guiding them through human-like logical reasoning processes.

Additionally, recent work explored the suitability of LLMs for recommenders, revealing that models, such as Llama and GPT, improve user satisfaction with richer explanations [33], excel in data-sparse [42] and cold-start [50] scenarios, and can deliver highly relevant recommendations through conversational interactions [37].

**This Work.** In this paper, we are—to the best of our knowledge—the first to address the suitability of LLMs as narrative-driven recommenders. In particular, we utilize user-submitted text from reddit, consisting of narrative requests for movie recommendations, to evaluate the quality of LLM-generated recommendations.

## 6 Conclusion

In this paper, we evaluate the suitability of LLMs in a narrative-driven movie recommendation setting, comparing their performance to traditional state-of-the-art recommender systems. Our comprehensive analysis of 38 LLMs, both open- and closed-source, reveals that LLMs can effectively generate personalized movie recommendations from user-provided narratives, significantly outperforming traditional approaches, such as doc2vec. We find that while larger closed-source models generally demonstrate superior performance, medium-sized open-source models remain competitive, offering a viable trade-off between computational cost and recommendation quality. Notably, we observe minimal differences in effectiveness between zero-shot, identity, and few-shot prompting, indicating that simple approaches are sufficient for generating high-quality recommendations. These findings highlight the potential of LLMs to transform narrative-driven recommender systems, offering scalable solutions for integrating natural language capabilities into real-world applications. Our results also underscore the versatility of LLMs in real-world recommender applications. Minimizing prompt complexity is crucial, as it reduces operational overhead and latency, simplifying the integration of LLMs into existing recommendation frameworks. This finding is particularly valuable for researchers and practitioners aiming to incorporate LLMs in practical settings without the additional computational costs of model fine-tuning or complex prompt engineering.

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

---

**Prompt for Labeling Robustness Analysis Dataset**

**Task**

Detect all movies that are mentioned in a user's request. The user request is given in """ tags.
Respond with a JSON object that contains a list of all mentioned movies, where each movie is provided in a single string which matches the format "<movie_title> - <IMDb_id>". Omit any additional text or output.

**Examples**

Here is an example:
"""Requesting beautiful, thoughtful, complex movies about the nature of consciousness and what it means to be alive.\nGhost in the Shell (1995) is one of my favourites and I love it. I also liked Upgrade and Ex Machina although they were significantly less hopeful."""
{"movies": ["Ghost in the Shell - tt0113568", "Upgrade - tt6499752", "Ex Machina - tt0470752"]}

**Figure 7: Prompt for Robustness Analysis Dataset Creation. We visualize the system prompt for tagging movies in reddit submissions for our robustness experiment addressing data leakage (Sec. 3.3). It contains the task and format constraints as well as a few-shot example. When prompting GPT-4o as an expert labeler, we utilize this system prompt along user prompts containing reddit submission texts.**

## A Appendix

We show supplementary materials and figures for multiple subsections of the main body of our paper in this Appendix.

**Evaluation of LLM Responses: Response Variance.** For our analysis estimating the variance of LLM responses (Sec. 2.3), we show detailed results of our one-way ANOVA in Table 2.

**Results: Format Adherence.** While we show aggregated results for all metrics measuring formatting and structural correctness of LLM responses in the main part of the paper (Sec. 3.1), we hereafter visualize detailed results for all models regarding valid JSON format (Fig. 8), number of recommendations (Fig. 9), unique movies (Fig. 10), and movies released before the request (Fig. 11).

**Results: Addressing Potential Data Leakage.** We show the prompt we utilize to label our robustness analysis dataset using GPT-4o in Figure 7.

**Results: Recommendation Performance.** In our main result section, we report aggregate performance metrics as well as discuss F1@10 across models in regard to their model families and size category (Sec. 3.2). To supplement this, we now plot results for Precision@10, Recall@10, and NDCG@10—which do not differ substantially to F1@10—in similar style (Figs. 13, 14, 15) as well as further detailing results for F1@10 in a heatmap (Fig. 12).

**Discussion: Diversity and Movie Years.** We substantiate the aggregated plot of diversity for all model size categories (Sec. 4, Fig. 5) by visualizing inter-list diversity for individual models in Fig. 16. Furthermore, we lay out more details about the release year distribution of the movies contained in LLM responses in Fig. 17.

**Table 2: Zero-Shot Performance Comparison Across 30 Repetitions Using One-Way ANOVA RM. This table presents the F-values (F Value) and associated p-values (Pr > F) for Precision@10, Recall@10, F1@10, and NDCG@10 metrics across 30 repetitions using zero-shot prompting for each LLM. The analysis is based on 29 numerator degrees of freedom (DF) and 8 555 denominator DF. The results exhibit no statistically significant differences in performance metrics between repetitions at the standard threshold of $p < 0.05$. To account for multiple tests we use Bonferroni correction and divide the significance level of 0.05 with the total number of tests (152) to obtain the final significance level of $p < 3.3 \cdot 10^{-4}$.**

| Model Category | Model Name | Precision@10 | | Recall@10 | | F1@10 | | NDCG@10 | |
|---|---|---|---|---|---|---|---|---|---|
| | | F Value | Pr > F | F Value | Pr > F | F Value | Pr > F | F Value | Pr > F |
| Tiny | Gemma 2B | 0.5892 | 0.9607 | 1.0027 | 0.4612 | 0.8696 | 0.6667 | 0.8290 | 0.7267 |
| | Gemma 2 2B | 0.5518 | 0.9754 | 0.7571 | 0.8218 | 0.6942 | 0.8885 | 0.5749 | 0.9669 |
| | Llama 3.2 1B | 1.5303 | 0.0341 | 1.7137 | 0.0099 | 1.6844 | 0.0122 | 1.5194 | 0.0366 |
| | Llama 3.2 3B | 1.1144 | 0.3066 | 1.2545 | 0.1633 | 1.1839 | 0.2278 | 1.1578 | 0.2556 |
| | Phi-3 3.8B | 1.0710 | 0.3630 | 0.6720 | 0.9079 | 0.7118 | 0.8716 | 0.9528 | 0.5377 |
| | Phi-3.5 3.8B | 0.9381 | 0.5607 | 1.1038 | 0.3199 | 1.0670 | 0.3684 | 0.9265 | 0.5788 |
| | Qwen2 0.5B | 0.8661 | 0.6720 | 0.8565 | 0.6865 | 0.8247 | 0.7328 | 0.8677 | 0.6696 |
| | Qwen2 1.5B | 0.7861 | 0.7855 | 1.3928 | 0.0782 | 1.2753 | 0.1472 | 0.9708 | 0.5099 |
| | Qwen2.5 0.5B | 1.3712 | 0.0884 | 0.9001 | 0.6199 | 0.9464 | 0.5477 | 1.3100 | 0.1231 |
| | Qwen2.5 1.5B | 0.8943 | 0.6289 | 1.0432 | 0.4017 | 1.0125 | 0.4466 | 0.8725 | 0.6623 |
| | Qwen2.5 3B | 0.7361 | 0.8460 | 0.7078 | 0.8756 | 0.7026 | 0.8806 | 0.7395 | 0.8422 |
| Small | Gemma 7B | 0.9148 | 0.5970 | 0.9425 | 0.5537 | 0.9248 | 0.5815 | 0.6911 | 0.8914 |
| | Gemma 2 9B | 1.2181 | 0.1946 | 1.3006 | 0.1293 | 1.2950 | 0.1331 | 1.0597 | 0.3785 |
| | GPT-4o mini | 1.0238 | 0.4299 | 0.8861 | 0.6416 | 0.9437 | 0.5519 | 1.3070 | 0.1251 |
| | Llama 3 8B | 1.0565 | 0.3829 | 0.8686 | 0.6683 | 0.8875 | 0.6393 | 1.0257 | 0.4271 |
| | Llama 3.1 8B | 0.9477 | 0.5457 | 0.7396 | 0.8421 | 0.7734 | 0.8017 | 0.7961 | 0.7723 |
| | Mistral 7B | 1.4327 | 0.0621 | 1.3772 | 0.0855 | 1.3807 | 0.0838 | 1.2420 | 0.1736 |
| | Qwen2 7B | 0.9254 | 0.5805 | 0.9736 | 0.5056 | 0.9290 | 0.5748 | 0.7936 | 0.7756 |
| | Qwen2.5 7B | 0.7321 | 0.8503 | 0.6830 | 0.8986 | 0.6897 | 0.8926 | 0.8768 | 0.6558 |
| | Yi 6B | 0.6908 | 0.8916 | 0.7751 | 0.7996 | 0.7841 | 0.7881 | 0.7409 | 0.8406 |
| | Yi 9B | 1.0973 | 0.3281 | 1.1183 | 0.3018 | 1.1393 | 0.2766 | 1.2357 | 0.1789 |
| Medium | Gemma 2 27B | 1.0867 | 0.3419 | 1.0807 | 0.3499 | 1.0892 | 0.3387 | 1.0701 | 0.3643 |
| | GPT-3.5 Turbo | 0.8981 | 0.6230 | 0.9619 | 0.5235 | 0.9313 | 0.5712 | 0.9267 | 0.5785 |
| | Mistral NeMo 12B | 0.8833 | 0.6459 | 0.8893 | 0.6367 | 0.8701 | 0.6660 | 0.7129 | 0.8704 |
| | Mistral Small 22B | 0.9447 | 0.5504 | 0.9341 | 0.5669 | 0.8949 | 0.6279 | 0.9096 | 0.6052 |
| | Mixtral 8x7B | 0.6807 | 0.9005 | 0.6872 | 0.8949 | 0.6802 | 0.9010 | 0.8734 | 0.6609 |
| | Phi-3 14B | 0.7426 | 0.8387 | 0.8350 | 0.7181 | 0.8038 | 0.7619 | 0.6936 | 0.8890 |
| | Qwen2.5 14B | 1.6317 | 0.0175 | 1.3929 | 0.0782 | 1.4445 | 0.0579 | 1.7059 | 0.0105 |
| | Qwen2.5 32B | 1.3921 | 0.0785 | 1.1496 | 0.2648 | 1.1724 | 0.2397 | 1.5363 | 0.0328 |
| | Yi 34B | 0.4645 | 0.9937 | 0.5696 | 0.9690 | 0.5277 | 0.9824 | 0.7070 | 0.8764 |
| Large | GPT-4o | 1.1677 | 0.2448 | 1.2949 | 0.1332 | 1.2904 | 0.1363 | 0.7112 | 0.8722 |
| | Llama 3 70B | 0.7460 | 0.8348 | 0.8824 | 0.6472 | 0.8725 | 0.6623 | 0.5791 | 0.9651 |
| | Llama 3.1 70B | 1.4396 | 0.0596 | 1.3234 | 0.1147 | 1.3243 | 0.1141 | 1.4188 | 0.0674 |
| | Llama 3.1 405B | 1.0021 | 0.4620 | 0.9738 | 0.5051 | 1.0004 | 0.4647 | 0.9988 | 0.4671 |
| | Mistral Large 123B | 1.1131 | 0.3082 | 1.0666 | 0.3690 | 1.0983 | 0.3269 | 1.4827 | 0.0459 |
| | Mixtral 8x22B | 0.9408 | 0.5565 | 0.7787 | 0.7951 | 0.8451 | 0.7033 | 1.0889 | 0.3391 |
| | Qwen2 72B | 0.8674 | 0.6701 | 0.7723 | 0.8032 | 0.7891 | 0.7816 | 0.8315 | 0.7231 |
| | Qwen2.5 72B | 1.0417 | 0.4039 | 1.0040 | 0.4592 | 1.0458 | 0.3980 | 1.1607 | 0.2525 |

**Valid JSON Format**

| | | Zero-Shot | Identity: Reddit User | Identity: Movie Critic | Identity: Movie Rec. | Few-Shot: 1 Example | Few-Shot: 5 Examples | Few-Shot: 10 Examples |
|---|---|---|---|---|---|---|---|---|
| Tiny | Gemma 2B | 1.0000 [1.0000, 1.0000] | 1.0000 [1.0000, 1.0000] | 1.0000 [1.0000, 1.0000] | 1.0000 [1.0000, 1.0000] | .9628 [.9426, .9865] | .9730 [.9561, .9932] | .9493 [.9257, .9764] |
| | Gemma 2 2B | .9797 [.9662, .9966] | .9628 [.9426, .9865] | .9797 [.9662, .9966] | .9730 [.9561, .9932] | .9966 [.9932, 1.0034] | .9932 [.9865, 1.0034] | .9797 [.9662, .9966] |
| | Llama 3.2 1B | .9291 [.9020, .9595] | .9223 [.8919, .9527] | .8953 [.8615, .9324] | .9054 [.8750, .9392] | .9426 [.9189, .9696] | .9628 [.9426, .9865] | .9797 [.9662, .9966] |
| | Llama 3.2 3B | .9966 [.9932, 1.0034] | 1.0000 [1.0000, 1.0000] | .9932 [.9865, 1.0034] | 1.0000 [1.0000, 1.0000] | .9966 [.9932, 1.0034] | .9966 [.9932, 1.0034] | .9899 [.9797, 1.0034] |
| | Phi-3 3.8B | .9797 [.9662, .9966] | .9932 [.9865, 1.0034] | .9831 [.9696, 1.0034] | .9966 [.9932, 1.0034] | .9932 [.9865, 1.0034] | .9932 [.9865, 1.0034] | .9831 [.9696, 1.0000] |
| | Phi-3.5 3.8B | .9966 [.9932, 1.0034] | 1.0000 [1.0000, 1.0000] | .9932 [.9865, 1.0034] | .9899 [.9797, 1.0034] | .9966 [.9932, 1.0034] | .9831 [.9696, 1.0000] | .9730 [.9561, .9932] |
| | Qwen2 0.5B | .8818 [.8446, .9189] | .8953 [.8615, .9324] | .8514 [.8108, .8919] | .8716 [.8345, .9122] | 1.0000 [1.0000, 1.0000] | .9966 [.9932, 1.0034] | .9966 [.9932, 1.0034] |
| | Qwen2 1.5B | .9966 [.9932, 1.0034] | .9899 [.9797, 1.0034] | 1.0000 [1.0000, 1.0000] | .9966 [.9932, 1.0034] | 1.0000 [1.0000, 1.0000] | 1.0000 [1.0000, 1.0000] | 1.0000 [1.0000, 1.0000] |
| | Qwen2.5 0.5B | .9797 [.9662, .9966] | .9865 [.9764, 1.0000] | .9932 [.9797, 1.0034] | .9932 [.9865, 1.0034] | 1.0000 [1.0000, 1.0000] | .9966 [.9932, 1.0034] | .9899 [.9797, 1.0034] |
| | Qwen2.5 1.5B | .9865 [.9764, 1.0000] | .9899 [.9797, 1.0034] | .9932 [.9865, 1.0034] | .9865 [.9764, 1.0000] | 1.0000 [1.0000, 1.0000] | 1.0000 [1.0000, 1.0000] | .9966 [.9932, 1.0034] |
| | Qwen2.5 3B | 1.0000 [1.0000, 1.0000] | .9966 [.9932, 1.0034] | 1.0000 [1.0000, 1.0000] | .9966 [.9932, 1.0034] | 1.0000 [1.0000, 1.0000] | 1.0000 [1.0000, 1.0000] | .9932 [.9865, 1.0034] |
| Small | Gemma 7B | 1.0000 [1.0000, 1.0000] | .9966 [.9932, 1.0034] | 1.0000 [1.0000, 1.0000] | 1.0000 [1.0000, 1.0000] | 1.0000 [1.0000, 1.0000] | 1.0000 [1.0000, 1.0000] | .9932 [.9865, 1.0034] |
| | Gemma 2 9B | 1.0000 [1.0000, 1.0000] | 1.0000 [1.0000, 1.0000] | 1.0000 [1.0000, 1.0000] | 1.0000 [1.0000, 1.0000] | 1.0000 [1.0000, 1.0000] | 1.0000 [1.0000, 1.0000] | 1.0000 [1.0000, 1.0000] |
| | GPT-4o mini | 1.0000 [1.0000, 1.0000] | 1.0000 [1.0000, 1.0000] | 1.0000 [1.0000, 1.0000] | 1.0000 [1.0000, 1.0000] | 1.0000 [1.0000, 1.0000] | 1.0000 [1.0000, 1.0000] | 1.0000 [1.0000, 1.0000] |
| | Llama 3 8B | .9966 [.9932, 1.0034] | .9966 [.9932, 1.0034] | 1.0000 [1.0000, 1.0000] | 1.0000 [1.0000, 1.0000] | 1.0000 [1.0000, 1.0000] | 1.0000 [1.0000, 1.0000] | .9932 [.9865, 1.0034] |
| | Llama 3.1 8B | 1.0000 [1.0000, 1.0000] | 1.0000 [1.0000, 1.0000] | 1.0000 [1.0000, 1.0000] | 1.0000 [1.0000, 1.0000] | 1.0000 [1.0000, 1.0000] | 1.0000 [1.0000, 1.0000] | 1.0000 [1.0000, 1.0000] |
| | Mistral 7B | .8243 [.7838, .8682] | .8682 [.8311, .9088] | .8209 [.7770, .8649] | .9155 [.8851, .9493] | 1.0000 [1.0000, 1.0000] | 1.0000 [1.0000, 1.0000] | .9932 [.9865, 1.0034] |
| | Qwen2 7B | 1.0000 [1.0000, 1.0000] | 1.0000 [1.0000, 1.0000] | .9932 [.9865, 1.0034] | 1.0000 [1.0000, 1.0000] | 1.0000 [1.0000, 1.0000] | 1.0000 [1.0000, 1.0000] | .9899 [.9797, 1.0034] |
| | Qwen2.5 7B | .9865 [.9764, 1.0000] | .9831 [.9696, 1.0000] | .9764 [.9595, .9932] | .9899 [.9797, 1.0034] | 1.0000 [1.0000, 1.0000] | 1.0000 [1.0000, 1.0000] | .9966 [.9932, 1.0034] |
| | Yi 6B | .9899 [.9797, 1.0034] | .9595 [.9392, .9831] | .8649 [.8277, .9054] | .9257 [.8986, .9561] | .9966 [.9932, 1.0034] | .9527 [.9291, .9797] | .9223 [.8919, .9527] |
| | Yi 9B | .9966 [.9932, 1.0034] | 1.0000 [1.0000, 1.0000] | 1.0000 [1.0000, 1.0000] | 1.0000 [1.0000, 1.0000] | .9966 [.9932, 1.0034] | 1.0000 [1.0000, 1.0000] | .9899 [.9797, 1.0034] |
| Medium | Gemma 2 27B | 1.0000 [1.0000, 1.0000] | 1.0000 [1.0000, 1.0000] | 1.0000 [1.0000, 1.0000] | .9966 [.9932, 1.0034] | 1.0000 [1.0000, 1.0000] | 1.0000 [1.0000, 1.0000] | .9932 [.9865, 1.0034] |
| | GPT-3.5 Turbo | 1.0000 [1.0000, 1.0000] | 1.0000 [1.0000, 1.0000] | 1.0000 [1.0000, 1.0000] | 1.0000 [1.0000, 1.0000] | 1.0000 [1.0000, 1.0000] | 1.0000 [1.0000, 1.0000] | 1.0000 [1.0000, 1.0000] |
| | Mistral NeMo 12B | 1.0000 [1.0000, 1.0000] | 1.0000 [1.0000, 1.0000] | 1.0000 [1.0000, 1.0000] | 1.0000 [1.0000, 1.0000] | 1.0000 [1.0000, 1.0000] | 1.0000 [1.0000, 1.0000] | .9966 [.9932, 1.0034] |
| | Mistral Small 22B | 1.0000 [1.0000, 1.0000] | 1.0000 [1.0000, 1.0000] | 1.0000 [1.0000, 1.0000] | 1.0000 [1.0000, 1.0000] | 1.0000 [1.0000, 1.0000] | 1.0000 [1.0000, 1.0000] | .9865 [.9764, 1.0000] |
| | Mixtral 8x7B | .9966 [.9932, 1.0034] | .9966 [.9932, 1.0034] | .9966 [.9932, 1.0034] | .9966 [.9932, 1.0034] | 1.0000 [1.0000, 1.0000] | .9966 [.9932, 1.0034] | .9966 [.9932, 1.0034] |
| | Phi-3 14B | 1.0000 [1.0000, 1.0000] | .9831 [.9696, 1.0000] | .9966 [.9932, 1.0034] | .9831 [.9696, 1.0000] | 1.0000 [1.0000, 1.0000] | 1.0000 [1.0000, 1.0000] | .6892 [.6385, .7432] |
| | Qwen2.5 14B | 1.0000 [1.0000, 1.0000] | 1.0000 [1.0000, 1.0000] | 1.0000 [1.0000, 1.0000] | 1.0000 [1.0000, 1.0000] | 1.0000 [1.0000, 1.0000] | 1.0000 [1.0000, 1.0000] | 1.0000 [1.0000, 1.0000] |
| | Qwen2.5 32B | 1.0000 [1.0000, 1.0000] | 1.0000 [1.0000, 1.0000] | 1.0000 [1.0000, 1.0000] | 1.0000 [1.0000, 1.0000] | 1.0000 [1.0000, 1.0000] | 1.0000 [1.0000, 1.0000] | .9966 [.9932, 1.0034] |
| | Yi 34B | 1.0000 [1.0000, 1.0000] | 1.0000 [1.0000, 1.0000] | 1.0000 [1.0000, 1.0000] | 1.0000 [1.0000, 1.0000] | .9899 [.9797, 1.0034] | .9797 [.9662, .9966] | .9324 [.9054, .9628] |
| Large | GPT-4o | 1.0000 [1.0000, 1.0000] | 1.0000 [1.0000, 1.0000] | 1.0000 [1.0000, 1.0000] | 1.0000 [1.0000, 1.0000] | 1.0000 [1.0000, 1.0000] | 1.0000 [1.0000, 1.0000] | 1.0000 [1.0000, 1.0000] |
| | Llama 3 70B | 1.0000 [1.0000, 1.0000] | 1.0000 [1.0000, 1.0000] | 1.0000 [1.0000, 1.0000] | 1.0000 [1.0000, 1.0000] | 1.0000 [1.0000, 1.0000] | 1.0000 [1.0000, 1.0000] | 1.0000 [1.0000, 1.0000] |
| | Llama 3.1 70B | 1.0000 [1.0000, 1.0000] | 1.0000 [1.0000, 1.0000] | 1.0000 [1.0000, 1.0000] | 1.0000 [1.0000, 1.0000] | 1.0000 [1.0000, 1.0000] | 1.0000 [1.0000, 1.0000] | 1.0000 [1.0000, 1.0000] |
| | Llama 3.1 405B | 1.0000 [1.0000, 1.0000] | 1.0000 [1.0000, 1.0000] | 1.0000 [1.0000, 1.0000] | 1.0000 [1.0000, 1.0000] | 1.0000 [1.0000, 1.0000] | .9966 [.9932, 1.0034] | 1.0000 [1.0000, 1.0000] |
| | Mistral Large 123B | 1.0000 [1.0000, 1.0000] | 1.0000 [1.0000, 1.0000] | 1.0000 [1.0000, 1.0000] | 1.0000 [1.0000, 1.0000] | 1.0000 [1.0000, 1.0000] | 1.0000 [1.0000, 1.0000] | .9899 [.9797, 1.0034] |
| | Mixtral 8x22B | 1.0000 [1.0000, 1.0000] | 1.0000 [1.0000, 1.0000] | 1.0000 [1.0000, 1.0000] | 1.0000 [1.0000, 1.0000] | 1.0000 [1.0000, 1.0000] | 1.0000 [1.0000, 1.0000] | .9932 [.9865, 1.0034] |
| | Qwen2 72B | 1.0000 [1.0000, 1.0000] | 1.0000 [1.0000, 1.0000] | 1.0000 [1.0000, 1.0000] | 1.0000 [1.0000, 1.0000] | 1.0000 [1.0000, 1.0000] | 1.0000 [1.0000, 1.0000] | .9966 [.9932, 1.0034] |
| | Qwen2.5 72B | 1.0000 [1.0000, 1.0000] | 1.0000 [1.0000, 1.0000] | 1.0000 [1.0000, 1.0000] | 1.0000 [1.0000, 1.0000] | 1.0000 [1.0000, 1.0000] | 1.0000 [1.0000, 1.0000] | .9966 [.9932, 1.0034] |

Scale: .7  .75  .8  .85  .9  .95  1

**Figure 8: JSON Format.** This figure shows the proportion of valid JSON responses generated by LLMs [bootstrapped 95% confidence intervals]. The results are categorized by model size and prompting strategies. Despite observing high proportion of valid output across all models, increased model size impacts the accuracy of valid output. Larger models achieve higher percentages of valid responses, demonstrating a benefit with scale.

Figure 9: Number of Recommendations. This figure presents the proportion of LLM-generated responses containing zero, one to nine, ten, and more than ten recommendations [bootstrapped 95% confidence intervals]. The results are categorized by model size and prompting strategies. Larger models consistently provide more accurate numbers of recommendations, adhering closely to the prompt specifications, while smaller models display greater variability.

| | | Zero-Shot | Identity: Reddit User | Identity: Movie Critic | Identity: Movie Rec. | Few-Shot: 1 Example | Few-Shot: 5 Examples | Few-Shot: 10 Examples |
|---|---|---|---|---|---|---|---|---|
| Tiny | Gemma 2B | .9745 [.9677, .9819] | .9736 [.9669, .9808] | .9781 [.9722, .9846] | .9753 [.9675, .9840] | .9629 [.9557, .9705] | .9610 [.9522, .9706] | .9529 [.9415, .9654] |
| | Gemma 2 2B | .9971 [.9952, .9996] | .9977 [.9961, .9996] | .9974 [.9955, .9995] | .9972 [.9953, .9994] | .9988 [.9976, 1.0006] | .9983 [.9971, 1.0000] | .9982 [.9968, 1.0001] |
| | Llama 3.2 1B | .9991 [.9983, 1.0009] | .9974 [.9948, 1.0014] | 1.0000 [1.0000, 1.0000] | .9947 [.9895, 1.0053] | 1.0000 [1.0000, 1.0000] | .9984 [.9968, 1.0016] | .9986 [.9971, 1.0011] |
| | Llama 3.2 3B | 1.0000 [1.0000, 1.0000] | 1.0000 [1.0000, 1.0000] | .9985 [.9973, 1.0000] | .9992 [.9984, 1.0004] | .9980 [.9964, 1.0005] | .9988 [.9977, 1.0002] | .9994 [.9989, 1.0003] |
| | Phi-3 3.8B | .9988 [.9975, 1.0004] | .9981 [.9965, .9999] | .9979 [.9962, 1.0000] | .9990 [.9980, 1.0007] | 1.0000 [1.0000, 1.0000] | 1.0000 [1.0000, 1.0000] | .9982 [.9964, 1.0006] |
| | Phi-3.5 3.8B | .9997 [.9994, 1.0003] | .9989 [.9979, 1.0003] | .9983 [.9970, 1.0001] | .9992 [.9985, 1.0004] | .9996 [.9991, 1.0004] | .9980 [.9959, 1.0020] | .9979 [.9959, 1.0010] |
| | Qwen2 0.5B | 1.0000 [1.0000, 1.0000] | 1.0000 [1.0000, 1.0000] | 1.0000 [1.0000, 1.0000] | .9667 [.9333, 1.0333] | 1.0000 [1.0000, 1.0000] | 1.0000 [1.0000, 1.0000] | .9978 [.9957, 1.0014] |
| | Qwen2 1.5B | .9912 [.9849, .9997] | .9897 [.9843, .9961] | .9933 [.9892, .9980] | .9846 [.9755, .9959] | .9961 [.9937, .9991] | .9962 [.9936, .9995] | .9971 [.9948, 1.0000] |
| | Qwen2.5 0.5B | .9532 [.9064, 1.0351] | .9800 [.9600, 1.0200] | .9859 [.9719, 1.0078] | .9890 [.9780, 1.0110] | .9911 [.9858, .9976] | .9919 [.9875, .9971] | .9919 [.9878, .9966] |
| | Qwen2.5 1.5B | .9893 [.9833, .9964] | .9972 [.9949, 1.0000] | .9856 [.9779, .9945] | .9811 [.9723, .9913] | .9952 [.9921, .9989] | .9929 [.9895, .9969] | .9911 [.9870, .9958] |
| | Qwen2.5 3B | .9937 [.9909, .9970] | .9902 [.9859, .9954] | .9949 [.9924, .9979] | .9911 [.9876, .9951] | .9907 [.9868, .9954] | .9877 [.9837, .9921] | .9873 [.9831, .9920] |
| Small | Gemma 7B | .9725 [.9658, .9799] | .9718 [.9648, .9795] | .9703 [.9639, .9774] | .9741 [.9680, .9806] | .9696 [.9617, .9783] | .9521 [.9426, .9622] | .9325 [.9176, .9488] |
| | Gemma 2 9B | .9986 [.9976, 1.0000] | 1.0000 [1.0000, 1.0000] | .9986 [.9976, 1.0000] | .9990 [.9980, 1.0007] | .9997 [.9993, 1.0003] | .9993 [.9986, 1.0003] | .9993 [.9986, 1.0007] |
| | GPT-4o mini | .9993 [.9986, 1.0003] | 1.0000 [1.0000, 1.0000] | 1.0000 [1.0000, 1.0000] | .9997 [.9993, 1.0003] | 1.0000 [1.0000, 1.0000] | .9997 [.9993, 1.0003] | 1.0000 [1.0000, 1.0000] |
| | Llama 3 8B | .9983 [.9969, 1.0003] | 1.0000 [1.0000, 1.0000] | .9993 [.9986, 1.0003] | .9990 [.9979, 1.0007] | 1.0000 [1.0000, 1.0000] | .9977 [.9961, .9995] | .9971 [.9952, .9995] |
| | Llama 3.1 8B | 1.0000 [1.0000, 1.0000] | .9996 [.9993, 1.0004] | .9997 [.9993, 1.0003] | .9990 [.9979, 1.0003] | .9997 [.9993, 1.0003] | .9989 [.9978, 1.0003] | .9983 [.9969, 1.0000] |
| | Mistral 7B | .9996 [.9991, 1.0004] | 1.0000 [1.0000, 1.0000] | 1.0000 [1.0000, 1.0000] | .9958 [.9927, 1.0008] | .9997 [.9993, 1.0003] | .9993 [.9986, 1.0003] | .9997 [.9993, 1.0003] |
| | Qwen2 7B | .9986 [.9975, 1.0000] | .9997 [.9993, 1.0003] | .9983 [.9965, 1.0009] | .9993 [.9986, 1.0003] | .9979 [.9962, 1.0000] | .9982 [.9967, .9999] | .9982 [.9968, 1.0004] |
| | Qwen2.5 7B | .9566 [.9424, .9722] | .9537 [.9388, .9702] | .9583 [.9439, .9742] | .9759 [.9662, .9871] | .9936 [.9905, .9973] | .9959 [.9936, .9986] | .9963 [.9942, .9986] |
| | Yi 6B | .9974 [.9954, 1.0000] | .9951 [.9908, 1.0007] | .9893 [.9821, .9987] | .9922 [.9862, .9999] | .9988 [.9978, 1.0002] | .9949 [.9915, .9997] | .9948 [.9917, .9991] |
| | Yi 9B | .9973 [.9955, .9993] | .9944 [.9903, .9997] | .9969 [.9951, .9990] | .9934 [.9902, .9971] | .9963 [.9942, .9986] | .9982 [.9968, .9998] | .9986 [.9975, 1.0000] |
| Medium | Gemma 2 27B | .9993 [.9986, 1.0003] | .9997 [.9993, 1.0003] | 1.0000 [1.0000, 1.0000] | 1.0000 [1.0000, 1.0000] | .9993 [.9986, 1.0003] | .9990 [.9980, 1.0003] | .9986 [.9976, 1.0000] |
| | GPT-3.5 Turbo | .9976 [.9959, .9997] | .9963 [.9939, .9993] | .9969 [.9953, .9990] | .9986 [.9976, 1.0000] | .9932 [.9899, .9970] | .9963 [.9943, .9986] | .9973 [.9956, .9993] |
| | Mistral NeMo 12B | .9993 [.9986, 1.0003] | .9993 [.9986, 1.0003] | 1.0000 [1.0000, 1.0000] | 1.0000 [1.0000, 1.0000] | .9986 [.9976, 1.0000] | .9997 [.9993, 1.0003] | .9997 [.9993, 1.0003] |
| | Mistral Small 22B | 1.0000 [1.0000, 1.0000] | .9976 [.9953, 1.0017] | .9993 [.9986, 1.0003] | .9993 [.9986, 1.0003] | 1.0000 [1.0000, 1.0000] | .9986 [.9976, 1.0000] | .9997 [.9993, 1.0003] |
| | Mixtral 8x7B | .9997 [.9993, 1.0003] | .9993 [.9986, 1.0003] | .9990 [.9980, 1.0003] | .9997 [.9993, 1.0003] | .9993 [.9986, 1.0003] | 1.0000 [1.0000, 1.0000] | .9990 [.9979, 1.0003] |
| | Phi-3 14B | .9986 [.9976, 1.0000] | .9992 [.9985, 1.0004] | .9986 [.9976, 1.0000] | .9979 [.9961, 1.0000] | .9958 [.9936, .9983] | .9948 [.9914, .9993] | .9929 [.9890, .9974] |
| | Qwen2.5 14B | .9990 [.9980, 1.0003] | .9993 [.9986, 1.0003] | .9986 [.9976, 1.0000] | .9997 [.9993, 1.0003] | .9997 [.9993, 1.0003] | 1.0000 [1.0000, 1.0000] | .9986 [.9976, 1.0000] |
| | Qwen2.5 32B | .9983 [.9970, 1.0000] | .9990 [.9980, 1.0003] | .9986 [.9976, 1.0000] | .9990 [.9980, 1.0003] | .9997 [.9993, 1.0003] | .9983 [.9970, 1.0000] | .9983 [.9969, 1.0000] |
| | Yi 34B | .9997 [.9993, 1.0003] | .9986 [.9973, 1.0010] | .9993 [.9986, 1.0003] | 1.0000 [1.0000, 1.0000] | .9997 [.9993, 1.0003] | 1.0000 [1.0000, 1.0000] | .9996 [.9993, 1.0004] |
| Large | GPT-4o | .9986 [.9976, 1.0000] | .9994 [.9987, 1.0003] | .9997 [.9993, 1.0003] | .9997 [.9993, 1.0003] | .9997 [.9993, 1.0003] | .9993 [.9986, 1.0003] | .9997 [.9993, 1.0003] |
| | Llama 3 70B | 1.0000 [1.0000, 1.0000] | 1.0000 [1.0000, 1.0000] | 1.0000 [1.0000, 1.0000] | 1.0000 [1.0000, 1.0000] | 1.0000 [1.0000, 1.0000] | 1.0000 [1.0000, 1.0000] | 1.0000 [1.0000, 1.0000] |
| | Llama 3.1 70B | 1.0000 [1.0000, 1.0000] | 1.0000 [1.0000, 1.0000] | 1.0000 [1.0000, 1.0000] | .9997 [.9993, 1.0003] | 1.0000 [1.0000, 1.0000] | 1.0000 [1.0000, 1.0000] | 1.0000 [1.0000, 1.0000] |
| | Llama 3.1 405B | 1.0000 [1.0000, 1.0000] | 1.0000 [1.0000, 1.0000] | 1.0000 [1.0000, 1.0000] | 1.0000 [1.0000, 1.0000] | 1.0000 [1.0000, 1.0000] | 1.0000 [1.0000, 1.0000] | 1.0000 [1.0000, 1.0000] |
| | Mistral Large 123B | 1.0000 [1.0000, 1.0000] | 1.0000 [1.0000, 1.0000] | 1.0000 [1.0000, 1.0000] | 1.0000 [1.0000, 1.0000] | 1.0000 [1.0000, 1.0000] | .9993 [.9986, 1.0003] | 1.0000 [1.0000, 1.0000] |
| | Mixtral 8x22B | .9997 [.9993, 1.0003] | 1.0000 [1.0000, 1.0000] | .9997 [.9993, 1.0003] | 1.0000 [1.0000, 1.0000] | 1.0000 [1.0000, 1.0000] | 1.0000 [1.0000, 1.0000] | 1.0000 [1.0000, 1.0000] |
| | Qwen2 72B | 1.0000 [1.0000, 1.0000] | 1.0000 [1.0000, 1.0000] | 1.0000 [1.0000, 1.0000] | 1.0000 [1.0000, 1.0000] | 1.0000 [1.0000, 1.0000] | 1.0000 [1.0000, 1.0000] | 1.0000 [1.0000, 1.0000] |
| | Qwen2.5 72B | .9993 [.9986, 1.0003] | .9997 [.9993, 1.0003] | 1.0000 [1.0000, 1.0000] | .9993 [.9986, 1.0003] | .9997 [.9993, 1.0003] | .9997 [.9993, 1.0003] | 1.0000 [1.0000, 1.0000] |

.93    .94    .95    .96    .97    .98    .99    1

**Figure 10: Unique Movies. This figure shows the average fraction of unique movie recommendations per request [bootstrapped 95% confidence intervals]. The results are categorized by model size and prompting strategies. Medium- and large-sized LLMs consistently returned over 99% unique movies within their valid JSON responses, while tiny and small models exhibited a slight but statistically significant increase in duplicate recommendations.**

**Movies Released Before Request**

| | | Zero-Shot | Identity: Reddit User | Identity: Movie Critic | Identity: Movie Rec. | Few-Shot: 1 Example | Few-Shot: 5 Examples | Few-Shot: 10 Examples |
|---|---|---|---|---|---|---|---|---|
| Tiny | Gemma 2B | .9353 [.9205,.9511] | .9225 [.9073,.9387] | .9076 [.8895,.9270] | .9109 [.8922,.9308] | .9183 [.9024,.9352] | .9067 [.8875,.9275] | .8989 [.8733,.9263] |
| | Gemma 2 2B | .9437 [.9329,.9553] | .9541 [.9446,.9642] | .9491 [.9384,.9605] | .9571 [.9481,.9667] | .9209 [.9001,.9444] | .9271 [.9132,.9417] | .9319 [.9194,.9453] |
| | Llama 3.2 1B | .9320 [.9054,.9629] | .9439 [.9215,.9699] | .9421 [.9250,.9606] | .9132 [.8809,.9502] | .9347 [.9187,.9526] | .9545 [.9425,.9676] | .9497 [.9367,.9641] |
| | Llama 3.2 3B | .9790 [.9721,.9868] | .9679 [.9558,.9822] | .9754 [.9665,.9855] | .9810 [.9734,.9904] | .9779 [.9710,.9856] | .9740 [.9664,.9824] | .9617 [.9497,.9753] |
| | Phi-3 3.8B | .9625 [.9518,.9752] | .9700 [.9614,.9795] | .9691 [.9610,.9781] | .9753 [.9677,.9840] | .9619 [.9510,.9742] | .9593 [.9480,.9721] | .9455 [.9315,.9607] |
| | Phi-3.5 3.8B | .9872 [.9821,.9932] | .9865 [.9822,.9913] | .9823 [.9764,.9889] | .9890 [.9846,.9941] | .9825 [.9768,.9887] | .9794 [.9723,.9873] | .9570 [.9460,.9687] |
| | Qwen2 0.5B | 1.0000 [1.0000,1.0000] | .8889 [.7778,1.1111] | .9343 [.8687,1.0404] | 1.0000 [1.0000,1.0000] | .9506 [.9189,.9906] | .9662 [.9477,.9889] | .9735 [.9607,.9882] |
| | Qwen2 1.5B | .9683 [.9552,.9854] | .9679 [.9541,.9850] | .9723 [.9605,.9870] | .9679 [.9541,.9842] | .9820 [.9740,.9912] | .9733 [.9616,.9874] | .9777 [.9664,.9914] |
| | Qwen2.5 0.5B | .7016 [.5386,.8858] | .8706 [.7611,1.0161] | .8797 [.7750,1.0250] | .8709 [.7844,.9740] | .9270 [.9082,.9479] | .9228 [.9058,.9413] | .9403 [.9251,.9569] |
| | Qwen2.5 1.5B | .9438 [.9224,.9682] | .9614 [.9475,.9776] | .9657 [.9535,.9800] | .9619 [.9494,.9759] | .9525 [.9393,.9674] | .9513 [.9399,.9637] | .9455 [.9340,.9577] |
| | Qwen2.5 3B | .9728 [.9655,.9807] | .9691 [.9609,.9781] | .9764 [.9697,.9837] | .9778 [.9710,.9853] | .9520 [.9414,.9632] | .9462 [.9340,.9596] | .9537 [.9436,.9647] |
| Small | Gemma 7B | .9389 [.9262,.9524] | .9492 [.9380,.9614] | .9459 [.9334,.9598] | .9484 [.9372,.9606] | .9468 [.9357,.9588] | .9255 [.9089,.9434] | .9462 [.9344,.9590] |
| | Gemma 2 9B | .9743 [.9676,.9818] | .9786 [.9725,.9852] | .9748 [.9676,.9826] | .9771 [.9713,.9833] | .9787 [.9719,.9868] | .9733 [.9666,.9811] | .9651 [.9570,.9739] |
| | GPT-4o mini | .9783 [.9726,.9845] | .9791 [.9733,.9855] | .9818 [.9767,.9872] | .9747 [.9682,.9818] | .9777 [.9716,.9845] | .9828 [.9777,.9885] | .9811 [.9757,.9868] |
| | Llama 3 8B | .9975 [.9958,.9995] | .9983 [.9969,1.0000] | .9983 [.9969,1.0000] | .9945 [.9917,.9976] | .9907 [.9852,.9986] | .9872 [.9828,.9920] | .9842 [.9790,.9900] |
| | Llama 3.1 8B | .9969 [.9951,.9993] | .9946 [.9911,.9993] | .9967 [.9944,.9993] | .9948 [.9921,.9983] | .9939 [.9905,.9976] | .9848 [.9797,.9905] | .9809 [.9748,.9876] |
| | Mistral 7B | .9966 [.9945,.9991] | .9975 [.9954,1.0000] | .9979 [.9963,.9999] | .9961 [.9938,.9987] | .9911 [.9877,.9948] | .9939 [.9911,.9969] | .9911 [.9873,.9952] |
| | Qwen2 7B | .9834 [.9769,.9908] | .9857 [.9806,.9912] | .9840 [.9786,.9900] | .9839 [.9774,.9917] | .9630 [.9529,.9747] | .9676 [.9582,.9783] | .9580 [.9486,.9681] |
| | Qwen2.5 7B | .9657 [.9565,.9757] | .9710 [.9626,.9807] | .9684 [.9592,.9787] | .9643 [.9529,.9778] | .9637 [.9532,.9756] | .9745 [.9681,.9814] | .9714 [.9642,.9793] |
| | Yi 6B | .9283 [.9099,.9489] | .9108 [.8867,.9374] | .8979 [.8689,.9299] | .9022 [.8765,.9305] | .9368 [.9231,.9514] | .9360 [.9220,.9510] | .9301 [.9127,.9492] |
| | Yi 9B | .9804 [.9744,.9871] | .9717 [.9620,.9828] | .9779 [.9706,.9864] | .9581 [.9440,.9741] | .9809 [.9741,.9891] | .9784 [.9724,.9850] | .9791 [.9723,.9869] |
| Medium | Gemma 2 27B | .9773 [.9712,.9837] | .9800 [.9746,.9861] | .9766 [.9708,.9827] | .9799 [.9745,.9857] | .9767 [.9706,.9831] | .9705 [.9634,.9783] | .9760 [.9704,.9820] |
| | GPT-3.5 Turbo | .9864 [.9813,.9920] | .9774 [.9682,.9882] | .9739 [.9634,.9868] | .9676 [.9557,.9814] | .9777 [.9686,.9889] | .9780 [.9709,.9858] | .9762 [.9684,.9860] |
| | Mistral NeMo 12B | .9964 [.9934,1.0000] | .9945 [.9903,1.0007] | .9965 [.9941,.9997] | .9935 [.9901,.9980] | .9959 [.9932,.9996] | .9949 [.9922,.9983] | .9885 [.9840,.9935] |
| | Mistral Small 22B | .9889 [.9828,.9966] | .9939 [.9909,.9973] | .9946 [.9915,.9980] | .9905 [.9861,.9956] | .9885 [.9845,.9929] | .9865 [.9807,.9936] | .9886 [.9838,.9948] |
| | Mixtral 8x7B | .9925 [.9892,.9963] | .9929 [.9898,.9963] | .9922 [.9888,.9959] | .9942 [.9919,.9969] | .9929 [.9878,1.0010] | .9884 [.9836,.9942] | .9894 [.9835,.9979] |
| | Phi-3 14B | .9900 [.9862,.9943] | .9902 [.9863,.9945] | .9857 [.9805,.9915] | .9878 [.9834,.9927] | .9876 [.9831,.9929] | .9678 [.9597,.9766] | .9697 [.9577,.9849] |
| | Qwen2.5 14B | .9838 [.9777,.9902] | .9824 [.9767,.9889] | .9817 [.9755,.9887] | .9806 [.9731,.9892] | .9830 [.9773,.9895] | .9774 [.9706,.9851] | .9834 [.9786,.9885] |
| | Qwen2.5 32B | .9919 [.9882,.9959] | .9924 [.9889,.9963] | .9922 [.9885,.9963] | .9965 [.9941,.9995] | .9929 [.9895,.9970] | .9909 [.9868,.9956] | .9891 [.9847,.9942] |
| | Yi 34B | .9728 [.9633,.9843] | .9770 [.9699,.9848] | .9768 [.9695,.9851] | .9838 [.9777,.9905] | .9784 [.9702,.9879] | .9723 [.9634,.9822] | .9746 [.9667,.9832] |
| Large | GPT-4o | .9963 [.9939,.9990] | .9963 [.9939,.9990] | .9963 [.9939,.9990] | .9959 [.9936,.9986] | .9959 [.9936,.9986] | .9956 [.9929,.9986] | .9966 [.9946,.9990] |
| | Llama 3 70B | .9941 [.9913,.9976] | .9965 [.9944,.9993] | .9982 [.9968,1.0000] | .9970 [.9951,.9997] | .9946 [.9919,.9980] | .9919 [.9882,.9959] | .9846 [.9798,.9901] |
| | Llama 3.1 70B | .9986 [.9973,1.0003] | .9993 [.9986,1.0003] | .9983 [.9970,1.0003] | .9969 [.9949,.9997] | .9969 [.9949,.9993] | .9929 [.9898,.9965] | .9892 [.9851,.9939] |
| | Llama 3.1 405B | .9986 [.9976,1.0000] | .9993 [.9986,1.0003] | .9993 [.9986,1.0004] | .9993 [.9986,1.0003] | 1.0000 [1.0000,1.0000] | 1.0000 [1.0000,1.0000] | .9980 [.9966,.9997] |
| | Mistral Large 123B | .9990 [.9980,1.0003] | .9980 [.9966,.9997] | .9993 [.9986,1.0003] | .9976 [.9959,.9997] | .9990 [.9980,1.0003] | .9986 [.9976,1.0000] | .9972 [.9955,.9993] |
| | Mixtral 8x22B | .9959 [.9936,.9986] | .9959 [.9936,.9986] | .9966 [.9945,.9990] | .9959 [.9939,.9983] | .9905 [.9868,.9949] | .9912 [.9875,.9953] | .9860 [.9798,.9942] |
| | Qwen2 72B | .9889 [.9843,.9943] | .9926 [.9892,.9963] | .9902 [.9861,.9949] | .9885 [.9841,.9936] | .9841 [.9787,.9902] | .9848 [.9797,.9905] | .9878 [.9831,.9932] |
| | Qwen2.5 72B | .9922 [.9885,.9970] | .9929 [.9892,.9973] | .9922 [.9889,.9963] | .9946 [.9919,.9980] | .9922 [.9889,.9959] | .9851 [.9804,.9905] | .9885 [.9844,.9929] |

.7   .75   .8   .85   .9   .95   1

**Figure 11: Movies Matching the Requested Year.** This figure presents the fraction of movies that correspond to the requested release year [bootstrapped 95% confidence intervals]. The results are categorized by model size and prompting strategies. The results highlight the ability of larger LLMs to comply with the constraints stated in the prompt. Tiny model, specifically Qwen2.5 0.5B with zero-shot prompting, struggled to recommend movies in the requested timespan with around 30% of the recommended movies outside the given period.

F1@10

| | | Zero-Shot | Identity: Reddit User | Identity: Movie Critic | Identity: Movie Rec. | Few-Shot: 1 Example | Few-Shot: 5 Examples | Few-Shot: 10 Examples |
|---|---|---|---|---|---|---|---|---|
| Tiny | Gemma 2B | .0325 [.0266, .0381] | .0325 [.0264, .0383] | .0340 [.0280, .0397] | .0335 [.0272, .0394] | .0307 [.0245, .0365] | .0245 [.0191, .0295] | .0170 [.0124, .0212] |
| | Gemma 2 2B | .0889 [.0783, .0993] | .1003 [.0884, .1116] | .0995 [.0880, .1105] | .0944 [.0827, .1056] | .0564 [.0465, .0658] | .0789 [.0682, .0891] | .0824 [.0717, .0926] |
| | Llama 3.2 1B | .0066 [.0037, .0091] | .0060 [.0030, .0086] | .0105 [.0067, .0139] | .0085 [.0047, .0118] | .0220 [.0165, .0272] | .0216 [.0165, .0264] | .0231 [.0179, .0279] |
| | Llama 3.2 3B | .0862 [.0744, .0975] | .0923 [.0804, .1037] | .0844 [.0744, .0943] | .0829 [.0725, .0931] | .1008 [.0895, .1117] | .0941 [.0831, .1047] | .0944 [.0838, .1047] |
| | Phi-3 3.8B | .0454 [.0377, .0526] | .0420 [.0351, .0485] | .0442 [.0370, .0511] | .0425 [.0357, .0491] | .0401 [.0339, .0461] | .0399 [.0333, .0462] | .0317 [.0256, .0375] |
| | Phi-3.5 3.8B | .0474 [.0397, .0547] | .0501 [.0422, .0576] | .0462 [.0387, .0534] | .0473 [.0391, .0549] | .0439 [.0364, .0511] | .0454 [.0376, .0529] | .0414 [.0344, .0481] |
| | Qwen2 0.5B | .0002 [−.0002, .0004] | .0000 [.0000, .0000] | .0000 [.0000, .0000] | .0000 [.0000, .0000] | .0004 [−.0003, .0008] | .0011 [−.0001, .0019] | .0012 [.0001, .0021] |
| | Qwen2 1.5B | .0083 [.0051, .0112] | .0125 [.0083, .0163] | .0139 [.0098, .0177] | .0119 [.0079, .0156] | .0183 [.0137, .0226] | .0198 [.0147, .0244] | .0205 [.0156, .0250] |
| | Qwen2.5 0.5B | .0000 [.0000, .0000] | .0000 [−.0003, .0006] | .0000 [.0000, .0000] | .0002 [−.0002, .0004] | .0034 [.0016, .0050] | .0037 [.0016, .0055] | .0032 [.0013, .0048] |
| | Qwen2.5 1.5B | .0124 [.0079, .0164] | .0125 [.0086, .0160] | .0091 [.0058, .0120] | .0112 [.0058, .0146] | .0199 [.0152, .0243] | .0212 [.0167, .0254] | .0246 [.0200, .0289] |
| | Qwen2.5 3B | .0275 [.0219, .0328] | .0303 [.0248, .0355] | .0258 [.0200, .0311] | .0300 [.0243, .0354] | .0420 [.0345, .0490] | .0413 [.0341, .0481] | .0388 [.0323, .0450] |
| Small | Gemma 7B | .0796 [.0700, .0889] | .0834 [.0733, .0933] | .0800 [.0711, .0887] | .0792 [.0695, .0885] | .0655 [.0564, .0744] | .0614 [.0526, .0698] | .0588 [.0499, .0673] |
| | Gemma 2 9B | .1596 [.1473, .1717] | .1638 [.1508, .1765] | .1527 [.1399, .1653] | .1576 [.1450, .1700] | .1562 [.1442, .1679] | .1569 [.1445, .1689] | .1555 [.1429, .1678] |
| | GPT-4o mini | .1798 [.1669, .1923] | .1817 [.1692, .1941] | .1781 [.1654, .1905] | .1792 [.1661, .1922] | .1776 [.1651, .1899] | .1798 [.1671, .1923] | .1784 [.1655, .1910] |
| | Llama 3 8B | .1019 [.0901, .1132] | .1071 [.0952, .1185] | .1053 [.0940, .1163] | .1080 [.0964, .1194] | .1109 [.0993, .1222] | .1193 [.1071, .1310] | .1175 [.1061, .1286] |
| | Llama 3.1 8B | .1090 [.0978, .1198] | .1068 [.0945, .1187] | .1110 [.0997, .1219] | .1061 [.0947, .1171] | .1058 [.0946, .1167] | .1104 [.0989, .1216] | .1113 [.0997, .1227] |
| | Mistral 7B | .1053 [.0929, .1174] | .1135 [.1004, .1260] | .1040 [.0913, .1163] | .1194 [.1071, .1314] | .1342 [.1219, .1462] | .1330 [.1210, .1446] | .1323 [.1200, .1443] |
| | Qwen 7B | .0832 [.0737, .0926] | .0885 [.0779, .0986] | .0856 [.0758, .0950] | .0836 [.0739, .0929] | .0868 [.0771, .0962] | .0874 [.0770, .0975] | .0847 [.0753, .0939] |
| | Qwen2.5 7B | .0580 [.0485, .0670] | .0498 [.0416, .0575] | .0538 [.0446, .0623] | .0539 [.0452, .0621] | .0707 [.0617, .0795] | .0747 [.0655, .0835] | .0709 [.0614, .0799] |
| | Yi 6B | .0230 [.0175, .0281] | .0191 [.0139, .0239] | .0206 [.0146, .0261] | .0201 [.0149, .0249] | .0380 [.0317, .0440] | .0410 [.0336, .0479] | .0417 [.0348, .0482] |
| | Yi 9B | .0503 [.0429, .0573] | .0524 [.0449, .0595] | .0531 [.0455, .0604] | .0539 [.0455, .0616] | .0523 [.0441, .0600] | .0525 [.0447, .0599] | .0530 [.0447, .0608] |
| Medium | Gemma 2 27B | .1839 [.1700, .1975] | .1815 [.1680, .1946] | .1855 [.1711, .1995] | .1791 [.1653, .1925] | .1803 [.1661, .1941] | .1834 [.1695, .1969] | .1838 [.1697, .1976] |
| | GPT-3.5 Turbo | .1962 [.1809, .2112] | .1944 [.1796, .2090] | .1884 [.1739, .2028] | .1873 [.1732, .2013] | .1936 [.1778, .2090] | .1964 [.1820, .2106] | .1957 [.1819, .2093] |
| | Mistral NeMo 12B | .1260 [.1138, .1379] | .1285 [.1161, .1405] | .1278 [.1158, .1396] | .1299 [.1176, .1418] | .1277 [.1160, .1391] | .1309 [.1196, .1420] | .1322 [.1206, .1436] |
| | Mistral Small 22B | .1607 [.1469, .1742] | .1718 [.1583, .1852] | .1662 [.1527, .1794] | .1656 [.1519, .1788] | .1654 [.1520, .1784] | .1642 [.1505, .1776] | .1617 [.1476, .1754] |
| | Mixtral 8x7B | .1475 [.1346, .1601] | .1519 [.1394, .1643] | .1476 [.1350, .1601] | .1517 [.1389, .1643] | .1416 [.1284, .1545] | .1503 [.1376, .1630] | .1519 [.1388, .1648] |
| | Phi-3 14B | .1092 [.0979, .1200] | .1088 [.0973, .1200] | .1132 [.1019, .1242] | .1080 [.0975, .1184] | .1122 [.1008, .1233] | .1149 [.1036, .1259] | .0768 [.0653, .0878] |
| | Qwen2.5 14B | .1138 [.1019, .1252] | .1178 [.1065, .1289] | .1161 [.1045, .1273] | .1133 [.1017, .1246] | .1194 [.1079, .1306] | .1236 [.1121, .1348] | .1173 [.1060, .1283] |
| | Qwen2.5 32B | .1267 [.1148, .1382] | .1264 [.1151, .1373] | .1300 [.1180, .1417] | .1276 [.1163, .1386] | .1216 [.1098, .1328] | .1281 [.1170, .1389] | .1288 [.1176, .1397] |
| | Yi 34B | .1107 [.1001, .1211] | .1170 [.1061, .1275] | .1125 [.1015, .1232] | .1207 [.1095, .1317] | .1191 [.1074, .1303] | .1212 [.1098, .1323] | .1166 [.1055, .1275] |
| Large | GPT-4o | .2134 [.1994, .2273] | .2143 [.1997, .2285] | .2157 [.2009, .2302] | .2138 [.1992, .2281] | .2111 [.1972, .2246] | .2144 [.1997, .2288] | .2135 [.1990, .2279] |
| | Llama 3 70B | .1412 [.1289, .1531] | .1383 [.1257, .1506] | .1353 [.1234, .1468] | .1394 [.1266, .1519] | .1513 [.1388, .1636] | .1526 [.1398, .1651] | .1597 [.1465, .1726] |
| | Llama 3.1 70B | .1488 [.1365, .1609] | .1463 [.1327, .1595] | .1508 [.1374, .1639] | .1505 [.1373, .1636] | .1499 [.1368, .1628] | .1494 [.1363, .1624] | .1507 [.1375, .1636] |
| | Llama 3.1 405B | .1526 [.1389, .1660] | .1630 [.1501, .1757] | .1538 [.1409, .1666] | .1534 [.1408, .1658] | .1554 [.1427, .1678] | .1557 [.1422, .1690] | .1591 [.1455, .1723] |
| | Mistral Large 123B | .2044 [.1912, .2174] | .2083 [.1934, .2229] | .2045 [.1895, .2193] | .2037 [.1892, .2178] | .2067 [.1924, .2209] | .2080 [.1936, .2220] | .1988 [.1846, .2128] |
| | Mixtral 8x22B | .1895 [.1754, .2034] | .1953 [.1806, .2098] | .1954 [.1805, .2099] | .1943 [.1796, .2089] | .1962 [.1825, .2096] | .1998 [.1856, .2138] | .1987 [.1844, .2130] |
| | Qwen2 72B | .1564 [.1438, .1686] | .1647 [.1517, .1777] | .1564 [.1431, .1691] | .1613 [.1485, .1737] | .1635 [.1506, .1760] | .1706 [.1579, .1829] | .1693 [.1569, .1816] |
| | Qwen2.5 72B | .1499 [.1376, .1618] | .1519 [.1392, .1641] | .1501 [.1377, .1622] | .1493 [.1371, .1612] | .1527 [.1405, .1646] | .1535 [.1411, .1657] | .1539 [.1414, .1663] |

0    .025    .05    .075    .1    .125    .15    .175    .2

**Figure 12: Detailed Recommendation Performance of Different LLMs and Prompting Strategies. We report F1@10 [bootstrapped 95% confidence intervals] for evaluated LLMs. The results are categorized by model size and prompting strategies.**

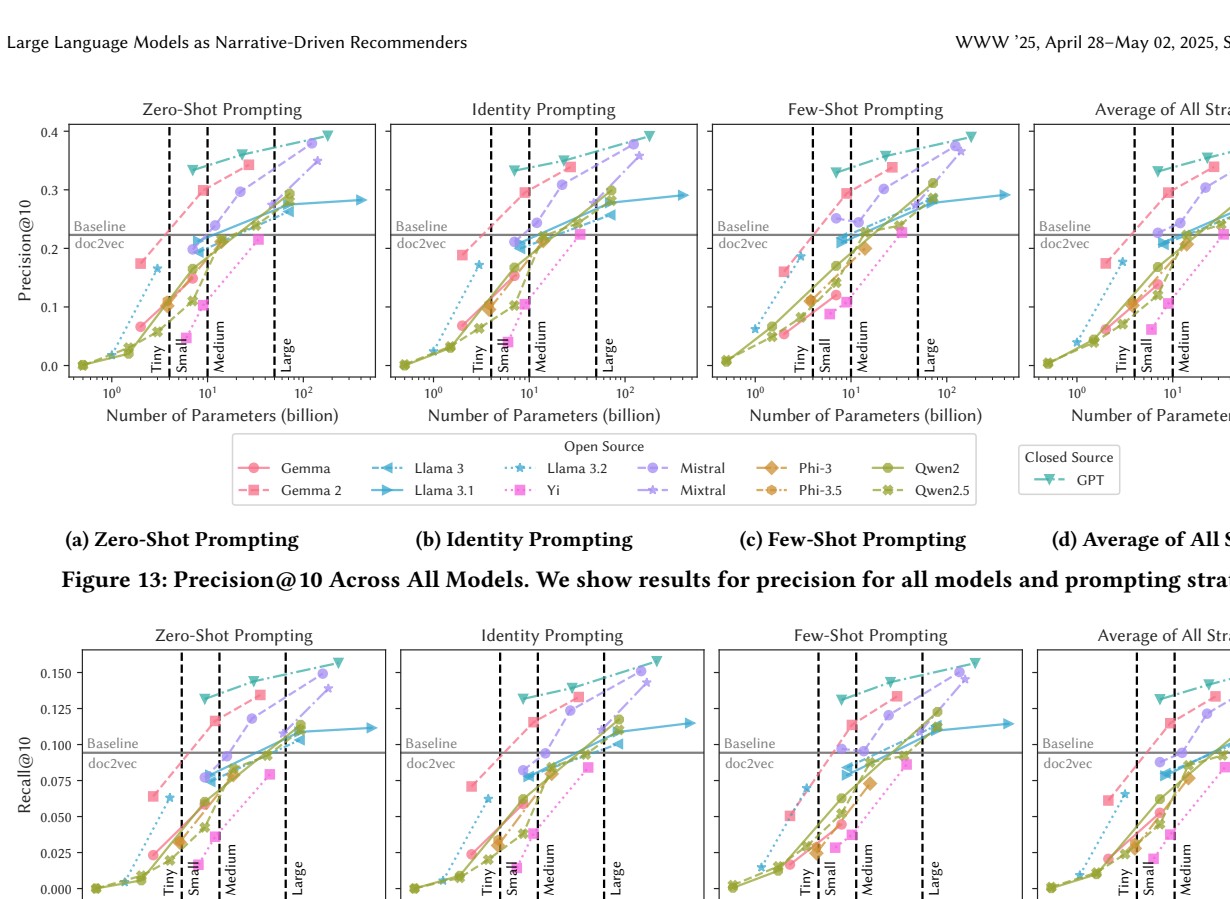

(a) Zero-Shot Prompting  (b) Identity Prompting  (c) Few-Shot Prompting  (d) Average of All Strategies

Figure 13: Precision@10 Across All Models. We show results for precision for all models and prompting strategies.

(a) Zero-Shot Prompting  (b) Identity Prompting  (c) Few-Shot Prompting  (d) Average of All Strategies

Figure 14: Recall@10 Across All Models. We show results for recall for all models and prompting strategies.

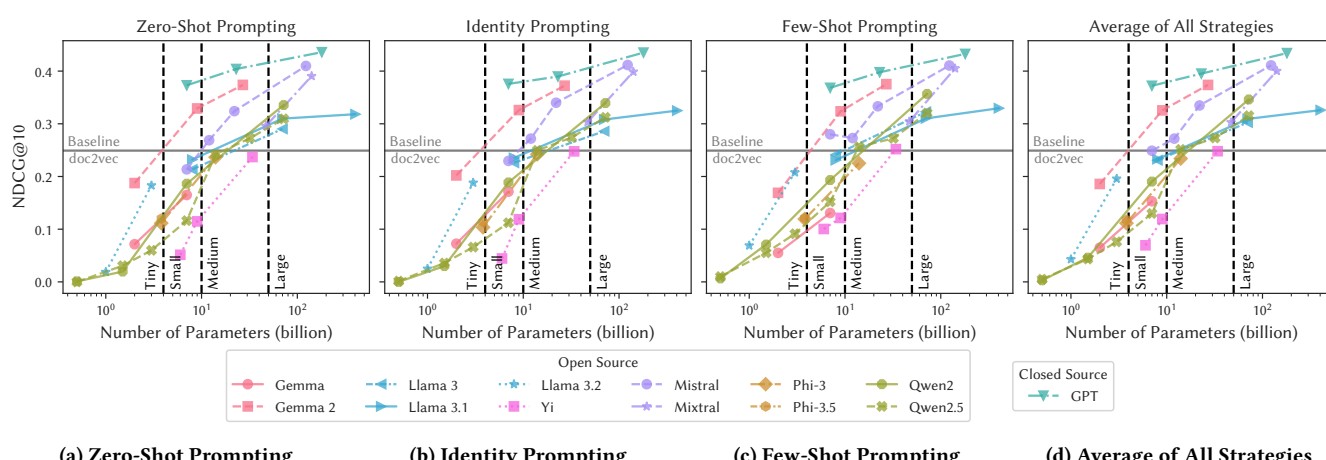

(a) Zero-Shot Prompting  (b) Identity Prompting  (c) Few-Shot Prompting  (d) Average of All Strategies

Figure 15: NDCG@10 Across All Models. We show results for NDCG for all models and prompting strategies.

Diversity@10

| | | Zero-Shot | Identity: Reddit User | Identity: Movie Critic | Identity: Movie Rec. | Few-Shot: 1 Example | Few-Shot: 5 Examples | Few-Shot: 10 Examples |
|---|---|---|---|---|---|---|---|---|
| Tiny | Gemma 2B | .7634 [.7452, .7821] | .7767 [.7604, .7939] | .7679 [.7467, .7900] | .7936 [.7760, .8120] | .8984 [.8874, .9097] | .9367 [.9279, .9461] | .9607 [.9533, .9687] |
| | Gemma 2 2B | .7126 [.6929, .7330] | .7193 [.7019, .7373] | .7150 [.6976, .7331] | .7263 [.7071, .7464] | .7811 [.7606, .8029] | .7770 [.7598, .7947] | .7865 [.7705, .8032] |
| | Llama 3.2 1B | .9407 [.9229, .9599] | .9718 [.9601, .9852] | .9605 [.9482, .9744] | .9508 [.9285, .9829] | .9638 [.9565, .9715] | .9607 [.9520, .9704] | .9600 [.9517, .9693] |
| | Llama 3.2 3B | .7611 [.7421, .7804] | .7729 [.7557, .7902] | .7885 [.7712, .8062] | .7742 [.7568, .7922] | .7705 [.7536, .7880] | .7688 [.7520, .7859] | .7932 [.7778, .8091] |
| | Phi-3 3.8B | .9265 [.9182, .9349] | .9293 [.9210, .9382] | .9264 [.9180, .9351] | .9222 [.9129, .9320] | .9362 [.9277, .9453] | .9296 [.9206, .9393] | .9361 [.9267, .9462] |
| | Phi-3.5 3.8B | .8787 [.8647, .8937] | .8862 [.8733, .9001] | .8778 [.8645, .8919] | .8792 [.8650, .8943] | .8960 [.8850, .9072] | .8933 [.8825, .9047] | .8899 [.8774, .9032] |
| | Qwen2 0.5B | 1.0000 [1.0000, 1.0000] | 1.0000 [1.0000, 1.0000] | 1.0000 [1.0000, 1.0000] | .8495 [.7199, 1.0116] | .9272 [.8800, .9850] | .9810 [.9702, .9941] | .9866 [.9808, .9937] |
| | Qwen2 1.5B | .8985 [.8742, .9255] | .8753 [.8547, .8969] | .8501 [.8231, .8791] | .8779 [.8543, .9032] | .9013 [.8833, .9209] | .9131 [.8967, .9314] | .9162 [.9013, .9328] |
| | Qwen2.5 0.5B | .6833 [.4167, 1.0333] | .8167 [.6458, 1.0667] | .8482 [.6964, 1.1339] | .9917 [.9835, 1.0083] | .9670 [.9569, .9798] | .9746 [.9680, .9828] | .9789 [.9739, .9844] |
| | Qwen2.5 1.5B | .9098 [.8898, .9323] | .8960 [.8738, .9206] | .8728 [.8486, .8991] | .9128 [.8952, .9324] | .9249 [.9133, .9377] | .9412 [.9341, .9485] | .9438 [.9367, .9514] |
| | Qwen2.5 3B | .8753 [.8610, .8906] | .8909 [.8778, .9048] | .8551 [.8366, .8751] | .8867 [.8723, .9024] | .9129 [.9036, .9227] | .9137 [.9042, .9238] | .9170 [.9078, .9264] |
| Small | Gemma 7B | .6479 [.6300, .6658] | .6478 [.6301, .6658] | .6456 [.6285, .6632] | .6479 [.6300, .6663] | .7442 [.7256, .7629] | .7702 [.7516, .7890] | .7758 [.7576, .7944] |
| | Gemma 2 9B | .6037 [.5870, .6205] | .6033 [.5861, .6209] | .6236 [.6069, .6405] | .6060 [.5891, .6232] | .6569 [.6410, .6729] | .6538 [.6379, .6694] | .6461 [.6298, .6626] |
| | GPT-4o mini | .5996 [.5819, .6174] | .5772 [.5602, .5943] | .6021 [.5850, .6194] | .5934 [.5758, .6111] | .6361 [.6206, .6519] | .6458 [.6304, .6613] | .6448 [.6300, .6597] |
| | Llama 3 8B | .7888 [.7731, .8049] | .7792 [.7636, .7952] | .7743 [.7588, .7900] | .7719 [.7565, .7877] | .7619 [.7476, .7765] | .7611 [.7462, .7764] | .7806 [.7655, .7962] |
| | Llama 3.1 8B | .7930 [.7791, .8072] | .7837 [.7685, .7990] | .7895 [.7746, .8047] | .7846 [.7706, .7988] | .7962 [.7821, .8105] | .7857 [.7712, .8005] | .7892 [.7740, .8048] |
| | Mistral 7B | .5993 [.5751, .6238] | .5958 [.5746, .6176] | .6116 [.5886, .6346] | .6007 [.5789, .6228] | .6791 [.6610, .6976] | .7229 [.7069, .7389] | .7411 [.7242, .7583] |
| | Qwen 7B | .7981 [.7808, .8162] | .7954 [.7794, .8119] | .7965 [.7800, .8135] | .8008 [.7834, .8187] | .7901 [.7736, .8073] | .8098 [.7948, .8251] | .8020 [.7864, .8178] |
| | Qwen2.5 7B | .8502 [.8342, .8666] | .8698 [.8546, .8856] | .8672 [.8534, .8820] | .8533 [.8368, .8710] | .7863 [.7699, .8030] | .7956 [.7801, .8116] | .8015 [.7863, .8171] |
| | Yi 6B | .9193 [.9051, .9352] | .8975 [.8773, .9204] | .9096 [.8931, .9276] | .9165 [.9033, .9304] | .9341 [.9253, .9435] | .9184 [.9057, .9326] | .9132 [.9012, .9262] |
| | Yi 9B | .9114 [.9019, .9214] | .9165 [.9075, .9259] | .9145 [.9062, .9231] | .9112 [.9022, .9203] | .9193 [.9107, .9282] | .9017 [.8926, .9111] | .8884 [.8759, .9022] |
| Medium | Gemma 2 27B | .4847 [.4653, .5039] | .4980 [.4780, .5180] | .4806 [.4610, .5003] | .5016 [.4810, .5223] | .5224 [.5042, .5406] | .5361 [.5174, .5547] | .5292 [.5106, .5477] |
| | GPT-3.5 Turbo | .6130 [.5962, .6299] | .5905 [.5722, .6090] | .6059 [.5877, .6242] | .5953 [.5776, .6132] | .6486 [.6306, .6669] | .6400 [.6232, .6570] | .6743 [.6585, .6899] |
| | Mistral NeMo 12B | .7312 [.7131, .7496] | .7312 [.7136, .7489] | .7391 [.7228, .7554] | .7273 [.7106, .7440] | .7536 [.7383, .7690] | .7715 [.7562, .7873] | .7622 [.7462, .7784] |
| | Mistral Small 22B | .6755 [.6587, .6927] | .6608 [.6434, .6784] | .6708 [.6538, .6881] | .6806 [.6638, .6974] | .7037 [.6873, .7204] | .7281 [.7127, .7436] | .7296 [.7145, .7450] |
| | Mixtral 8x7B | .5378 [.5151, .5607] | .5433 [.5209, .5657] | .5406 [.5187, .5626] | .5302 [.5075, .5531] | .7194 [.6999, .7395] | .6640 [.6433, .6851] | .6600 [.6395, .6806] |
| | Phi-3 14B | .8135 [.7990, .8284] | .8225 [.8101, .8351] | .8058 [.7921, .8199] | .8008 [.7874, .8147] | .8146 [.8016, .8279] | .8237 [.8104, .8372] | .8550 [.8394, .8710] |
| | Qwen2.5 14B | .7105 [.6928, .7284] | .7174 [.6987, .7366] | .7080 [.6883, .7281] | .7144 [.6955, .7336] | .7501 [.7335, .7670] | .7355 [.7195, .7517] | .7536 [.7373, .7701] |
| | Qwen2.5 32B | .6783 [.6599, .6972] | .6780 [.6600, .6962] | .6796 [.6616, .6979] | .6797 [.6611, .6985] | .7033 [.6848, .7220] | .6928 [.6763, .7094] | .6921 [.6753, .7093] |
| | Yi 34B | .7597 [.7397, .7802] | .7402 [.7216, .7593] | .7466 [.7283, .7654] | .7402 [.7214, .7594] | .7732 [.7568, .7903] | .7424 [.7252, .7601] | .7334 [.7148, .7524] |
| Large | GPT-4o | .5859 [.5684, .6036] | .5724 [.5552, .5896] | .5796 [.5626, .5966] | .5837 [.5664, .6011] | .6274 [.6111, .6438] | .6274 [.6113, .6437] | .6300 [.6154, .6447] |
| | Llama 3 70B | .6006 [.5818, .6195] | .6029 [.5835, .6222] | .6027 [.5824, .6227] | .5932 [.5745, .6119] | .6023 [.5846, .6199] | .6077 [.5888, .6265] | .5970 [.5789, .6150] |
| | Llama 3.1 70B | .6543 [.6362, .6725] | .6596 [.6411, .6781] | .6515 [.6341, .6689] | .6578 [.6406, .6751] | .6349 [.6162, .6537] | .6383 [.6212, .6556] | .6567 [.6382, .6753] |
| | Llama 3.1 405B | .5902 [.5716, .6088] | .5759 [.5580, .5937] | .6012 [.5824, .6201] | .5929 [.5726, .6133] | .5900 [.5717, .6085] | .5848 [.5669, .6030] | .5813 [.5636, .5992] |
| | Mistral Large 123B | .5402 [.5209, .5596] | .5439 [.5248, .5631] | .5432 [.5241, .5627] | .5352 [.5167, .5538] | .5638 [.5461, .5815] | .5965 [.5796, .6136] | .6274 [.6100, .6449] |
| | Mixtral 8x22B | .5686 [.5483, .5889] | .5511 [.5312, .5713] | .5505 [.5315, .5695] | .5618 [.5415, .5821] | .6131 [.5953, .6309] | .6265 [.6091, .6441] | .6317 [.6136, .6499] |
| | Qwen2 72B | .6132 [.5946, .6320] | .6055 [.5882, .6234] | .6148 [.5960, .6337] | .6164 [.5975, .6357] | .6324 [.6145, .6504] | .6290 [.6127, .6454] | .6180 [.6012, .6351] |
| | Qwen2.5 72B | .6022 [.5831, .6219] | .6056 [.5869, .6245] | .5998 [.5810, .6187] | .5983 [.5793, .6173] | .6243 [.6064, .6427] | .6134 [.5955, .6313] | .6139 [.5972, .6306] |

.5    .6    .7    .8    .9    1

Figure 16: Inter-list Diversity Across Repetitions. The figure illustrates the inter-list Diversity@10 [bootstrapped 95% confidence intervals] of recommendations generated by different LLMs across 30 repetitions for zero-shot and three repetitions for other prompting strategies, showing the variability in the uniqueness of recommended items across runs.

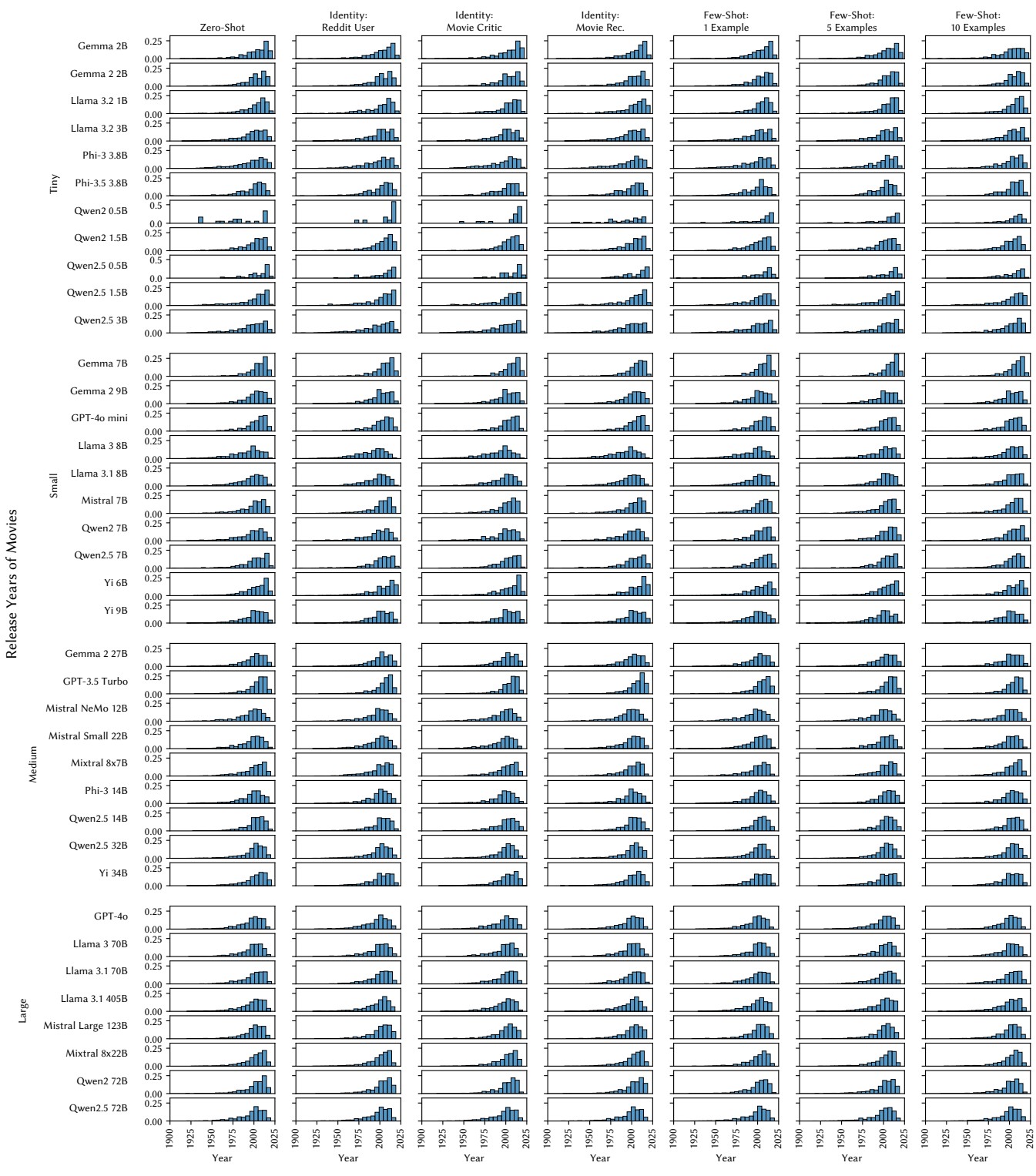

**Figure 17: Distribution of Movie Release Years. This figure shows the distribution of movies recommended by LLMs based on their release years. The results are categorized by model size and prompting strategies. The x-axis represents the year of release, spanning from 1950 to 2025, while the y-axis indicates the fraction of movies released each year. The data, aggregated in 5-year-spans, exhibits a noticeable increase in movie releases over time, with a sharp rise in the number of releases after the year 2000.**

