# OpenReview forum: "Large Language Models as Narrative-Driven Recommenders"
_ACM.org/TheWebConf/2025/Conference — WWW 2025 Poster_

### Official Review · Reviewer_z6zk · 2024-12-01

**Novelty:** 3
**Technical Quality:** 3

**Review:**

The paper evaluates open and closed-source LLMs across different parameter sizes for narrative recommendation, specifically focusing on a movie recommendation scenario using a dataset created by collecting movie request posts from Reddit communities. The experiments find that LLMs, especially larger ones, outperform the doc2vec model in narrative recommendation. Additionally, zero-shot prompting proves to be effective, while more complex prompting strategies do not significantly improve performance.

Strengths:
1. The evaluation includes a diverse range of LLMs with varying parameter sizes, making the findings valuable for the recommendation community.
2. The paper is easy to follow, with insightful discussions provided.

Weaknesses:
1. The finding that LLMs perform well in zero-shot narrative recommendation is only moderately novel. Given that LLMs inherently good at processing natural language queries without context, their strong performance in this zero-shot setting is somewhat expected.
2. As an evaluation paper, assessing narrative recommendation performance on only one dataset in a single domain may not provide sufficiently convincing evidence of generalizability.
3. The evaluation relies on ground truth extracted from community-provided comments. However, it is unclear whether these comments accurately fulfill the user requests, raising questions about the validity of evaluating model responses based on their alignment with these comments.
4. As noted by the authors, the LLMs were trained on publicly available corpora, and Reddit data may be included in this training. This could be a significant factor contributing to the better alignment of model responses with user comments.

**Questions:**

In Section 3.3, Addressing Potential Data Leakage, is any knowledge cutoff strategy used for testing models other than GPT-3.5 Turbo? How do they perform under this setup?

**Reviewer Confidence:**

4: The reviewer is certain that the evaluation is correct and very familiar with the relevant literature

**Scope:**

4: The work is relevant to the Web and to the track, and is of broad interest to the community

---

### Official Review · Reviewer_4Y4a · 2024-12-02

**Novelty:** 4
**Technical Quality:** 5

**Review:**

Summary:

In this paper, the authors applied Large Language Models (LLMs) to the narrative-driven recommender task, which aims to provide personalized suggestions based on user requests expressed in free-form text.
The authors conducted an evaluation of 38 state-of-the-art LLMs with diverse parameter scales, emphasizing the role of prompt engineering in enhancing performance within a movie recommendation context. To achieve this, they leveraged a gold-standard dataset annotated by crowdworkers, sourced from Reddit's movie suggestion community, and applied a range of prompting strategies, including zero-shot, identity-based, and few-shot approaches.

Strengths:

1.The experiments are extensive, including the performance evaluation of both open- and closed-source LLM across various parameter scales, the performance of different prompting strategies, and a comparison between traditional and state-of-the-art recommender approaches.
2.The experimental design demonstrates a high level of rigor, carefully accounting for potential challenges like data leakage and response variance to ensure reliability.
3. The authors have provided the source codes, which are good for the reproducibility of the experiments.

Weakness and Questions:

1.The output of the LLM is limited to a simple list containing only movie titles and release years. I am interested in understanding the specific reasons behind the LLM's recommendation of each movie and whether it can rank the movies by priority, which would help assess whether the LLM truly understands the task.
2.Does the hallucination issue of the LLM pose significant concerns? What is the probability of each model generating fictitious movies or incorrect information?
3.The User Prompt currently relies on free-form user requests, which introduces significant variability within the prompt structure. Have you considered leveraging the LLM to first analyze and summarize these requests, and then integrate the results into a structured prompt? This approach might further enhance the standardization and clarity of the overall prompt.

**Questions:**

1.The output of the LLM is limited to a simple list containing only movie titles and release years. I am interested in understanding the specific reasons behind the LLM's recommendation of each movie and whether it can rank the movies by priority, which would help assess whether the LLM truly understands the task.
2.Does the hallucination issue of the LLM pose significant concerns? What is the probability of each model generating fictitious movies or incorrect information?
3.The User Prompt currently relies on free-form user requests, which introduces significant variability within the prompt structure. Have you considered leveraging the LLM to first analyze and summarize these requests, and then integrate the results into a structured prompt? This approach might further enhance the standardization and clarity of the overall prompt.

**Reviewer Confidence:**

3: The reviewer is confident but not certain that the evaluation is correct

**Scope:**

3: The work is somewhat relevant to the Web and to the track, and is of narrow interest to a sub-community

---

### Official Review · Reviewer_sssJ · 2024-12-02

**Novelty:** 5
**Technical Quality:** 4

**Review:**

This paper presents a comprehensive evaluation of the effectiveness of large language models in generating personalized movie recommendations based on user-generated narrative requests. The authors compare the performance of 38 different LLMs, both open-source and closed-source, using a dataset derived from the MovieSuggestions subreddit. The study employs various prompting strategies and benchmarks the LLMs against traditional recommendation systems like doc2vec.

Pros:
1 By testing 38 different LLMs across various sizes and prompting strategies, the research provides a robust analysis of model performance, contributing valuable insights to the field.

Cons:
1. The study focuses solely on movie recommendations from a single subreddit, which may limit the generalizability of the findings to other domains.
2. The study does not explore more sophisticated prompting strategies (e.g., chain-of-thought prompting), which could enhance the adaptability and performance of the models.

**Questions:**

The same as above.

**Reviewer Confidence:**

3: The reviewer is confident but not certain that the evaluation is correct

**Scope:**

3: The work is somewhat relevant to the Web and to the track, and is of narrow interest to a sub-community

---

### Official Review · Reviewer_Xy5E · 2024-12-02

**Novelty:** 4
**Technical Quality:** 6

**Review:**

The paper presents a well documented, thorough analysis of LLM usage for generating recommendations given natural-language narratives, as opposed to the perhaps simpler cases of recommendations given other item scores (=classic recommender systems) or given queries (=search). To perform this analysis, the authors take an existing dataset of actual online conversations from Reddit which use a format of a natural language ask and several human recommendations that follow it. The different methods are then compared to the human recommendations. This is a very promising and worthy direction of research, and the presented study represents a significant amount of work. I think it is important to have such comparison studies, and liked the depth and breadth of the analysis. I was also happy to see the authors include an analysis of possible data leakage impact on the work's results.
Having said that, I have two major concerns for this paper:
1. Unsure if there is much novelty in using an LLM to solve a specific existing problem better than pre-LLM models.
2. I am missing at least a small attempt at human analysis of the LLM recommendations. The shown precision is just for the movies that happen to be the same as the ones users on Reddit recommended. Since this is an open problem, this means that even the best precision statistics shown are under 40% - which is not a very good recommendation performance (a user talking to a system that is only right 40% of the time might be upset). I would have liked to see a small-scale precision study based on human opinion of the LLM recommendations - for example, by asking annotators to evaluate which of two recommendation lists seems more relevant to the input.

**Questions:**

1. When describing "duplicate recommendations" in Unique Movies section, it is a little ambiguous if the meaning is duplicates in the same recommendation (that is, responding given a single input with a list that contains movie A twice), or duplicate recommendations across the entire dataset (that is, recommending the movie A for two different inputs). Assuming it's the first, it would also be very interesting to know how diverse are the movie recommendations for different inputs, do LLMs over-recommend popular movies (as I would suspect they do), and if the different models exhibit different properties in this aspect.
2. What was your prompt engineering process, and did you try some accommodations of the prompts per model and/or per prompting method? It is quite possible that the smaller model performance could be improved by further by relatively simple techniques, and of course for the open models - by training.
3. Another analysis I am missing is hallucination - how often do the different LLMs recommend movies that do not actually exist?

**Reviewer Confidence:**

3: The reviewer is confident but not certain that the evaluation is correct

**Scope:**

4: The work is relevant to the Web and to the track, and is of broad interest to the community

---

### Official Review · Reviewer_vsna · 2024-12-03

**Novelty:** 4
**Technical Quality:** 5

**Review:**

**Summary**

This paper evaluates the suitability of LLMs in a narrative-driven movie recommendation setting, comparing their performance to traditional state-of-the-art RSs. The comprehensive analysis of 38 LLMs, both open- and closed-source, reveals that LLMs can effectively generate personalized movie recommendations from user-provided narratives, significantly outperforming
traditional approaches, such as doc2vec. Accordingly, four key findings are gained in this study, for instance, while larger closed-source models generally demonstrate superior performance, medium-sized open-source models remain competitive, offering a viable trade-off between computational cost and recommendation quality, and minimal differences in effectiveness are observed between zero-shot, identity, and few-shot prompting, indicating that simple approaches are sufficient for generating high-quality recommendations, etc. These findings offer practical insights for recommender system researchers and practitioners looking to integrate LLMs into real-world recommendation applications.

**Pros**

P1. The paper is well-written and organized, which makes it easy to follow.

P2. Extensive LLM-based methods with different model sizes and prompting strategies have been tested in the movie recommendation domain.

P3. Four key findings in the paper offer practical insights for recommender system researchers and practitioners looking to integrate LLMs into real-world recommendation applications.

P4. The source code is available for reproduction.

**Cons**

C1. I totally acknowledge the merits of this paper and the great efforts made by the authors to comprehensively compare different LLM-based models with different sizes and prompting strategies, whereby four key findings are obtained. In this sense, the technical contribution of this paper seems to be limited. I personally believe it may be more suitable for other tracks, such as the resource/survey/reproducibility track instead of the main track.

C2. Only one domain, i.e., movie recommendation is tested, which may limit the generalizability of the obtained findings in this paper, although the authors point out this is one of the limitations of their study.

C3. Some of the detailed settings are not clear to me (maybe I missed something), for instance, what if the recommendation results are missing in the response? Will their Precision/Recall values be set as 0? Besides, what if the number of recommendation results is missing? It seems that I see two related claims in the paper, where one is we will make further requests until we get exactly 10 recommendations, and the other is we will exclude recommendations with less than 10 items. I do not know which one to stick to in this paper.  The authors may need to explain these points in the setup part.

C4. The baseline methods seem to be quite old and naive, e.g., doc2vec, mf. Are there any deep learning-based methods, such as seq2seq models, etc.?

C5. How to select the few-shot examples? Will all the methods use the same sampled few-shot examples for a given input? Besides, the authors find that few-shot examples may not necessarily improve the performance compared with zero-shot. Is there any possibility that the sampled examples are not effective enough? If changing a more efficient sample strategy, will this affect the results? Besides, does the number of few-shot examples matter?

**Questions:**

Please refer to C3-C5

**Reviewer Confidence:**

4: The reviewer is certain that the evaluation is correct and very familiar with the relevant literature

**Scope:**

3: The work is somewhat relevant to the Web and to the track, and is of narrow interest to a sub-community